# AMDCP: Adaptive Mixture Density for Conformal Prediction

## Abstract

We present Adaptive Mixture Density for Conformal Prediction (AMDCP), a "glass-box" component-aware framework that co-designs the non-conformity scoring framework with a mixture-density head. Our weight-aware score yields analytic non-contiguous unions of ellipsoids with constant-time membership tests, preserving classical coverage guarantees while improving efficiency. A comprehensive empirical evaluation on real-world datasets from demand forecasting to protein property prediction demonstrates that AMDCP's regions are sharper than existing generative methods, while also being an order of magnitude faster at inference time and 2-3x faster in training. We complement these results with theory (finite-sample marginal validity; an asymptotic optimality guarantee; and approximate group-conditional coverage) and extensive ablations under extreme distributions, model misspecification, model backbones, and more. AMDCP turns CP into a practical tool for real-world implementation: it is valid by construction, produces shape-adaptive sharp predictive sets, and is systems-efficient for modern pipelines.

## 1 Introduction

Conformal Prediction (CP) has emerged as a powerful, distribution-free framework for uncertainty quantification, offering rigorous coverage guarantees without restrictive assumptions on the underlying data distribution (Vovk, 2012; Shafer & Vovk, 2008; Angelopoulos & Bates, 2021). This makes CP particularly attractive for high-stakes applications where reliable error control is paramount. However, a fundamental challenge persists: many CP methods struggle to produce efficient (i.e., narrow) prediction regions when confronted with complex conditional distributions ubiquitous in real-world scenarios. Intricate local data manifold structures often lead to overly conservative intervals that, while valid, may lack practical value by encompassing an unnecessarily large set of predictions.

While various approaches have sought to enhance CP's adaptivity, for instance, by incorporating local weighting schemes or attempting to model conditional quantiles (Tibshirani et al., 2019; Romano et al., 2019; Guan, 2020), a critical limitation often remains in their capacity to fully exploit highly flexible density models to construct prediction regions that truly mirror intricate local data characteristics. Existing methods often rely on unimodal or homoskedastic assumptions within their non-conformity scores or underlying predictive models, which fundamentally restricts their ability to adapt to more complex distributional shapes. Oftentimes, however, this fundamental problem with CP means that these algorithms often produce prediction regions of suboptimal volume when confronted with heteroskedastic noise, or complex conditional distributions—characteristics ubiquitous in real-world applications ranging from healthcare to autonomous systems. This gap is particularly pronounced in domains where predictive distributions must accurately capture inherent stochasticity and potential multimodality to be truly informative, all the while remaining fast at inference-time to enable real-world applications.

To address these limitations, we introduce Adaptive Mixture Density for Conformal Prediction (AMDCP), a novel methodology that integrates the expressive capacity of Mixture Density Networks (MDNs) (Bishop, 1994) within the CP framework. AMDCP adaptively models the conditional distribution $p(y|x)$ as a mixture of Gaussians, where the mixture parameters (weights, means, and covariances) are themselves functions of the input $x$, learned by a neural network.

The MDN is a universal approximator that has proven highly effective for modeling complex, real-world distributions across a wide array of demanding fields, with applications ranging from robotics

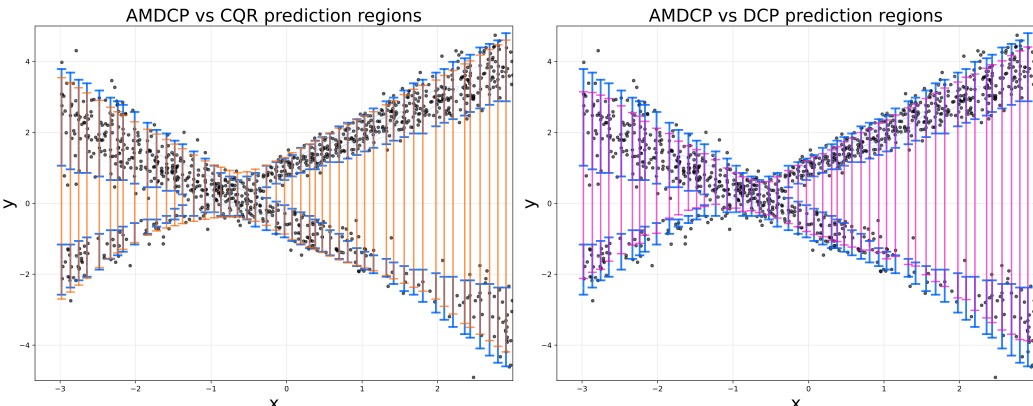

Figure 1: Visualization of non-contiguous prediction regions generated by AMDCP (blue) compared to CQR (orange) and an existing MDN-based method DCP (pink), for a synthetic dataset with bimodal, heteroskedastic noise at 90% coverage. AMDCP adapts to local data structure and its analytic union-of-ellipsoid method produces multiple disjoint prediction regions when appropriate. This gives the same coverage guarantee with narrower average prediction region volume.

and autonomous driving (Choi et al., 2018; Rehder et al., 2018), to generative speech synthesis (Zen & Senior, 2014), astrophysics (Teixeira et al., 2024), and critical healthcare tasks such as survival analysis (Han et al., 2022) and stochastic disease modeling (Carruthers & Finnie, 2023).

Crucially, we design a novel non-conformity score tailored to these mixture models. This allows AMDCP to construct prediction regions that automatically adapt their shape, orientation, and can even capture multimodality through potentially non-contiguous regions (as visualized in Figure 1, depicting disjoint prediction regions for bimodal data). This dynamic adaptation ensures that the prediction regions are both statistically valid, due to the CP framework, and highly efficient, by closely conforming to the local data density.

## 2 RELATED WORK, COMPARISON TO EXISTING METHODS, AND NOVELTY

The pursuit of reliable uncertainty quantification is a cornerstone of modern machine learning. Conformal prediction (CP) has emerged as a uniquely powerful framework for this task, offering rigorous, finite-sample coverage guarantees without imposing strong assumptions on the data-generating process or the underlying model (Vovk et al., 2005; Shafer & Vovk, 2008). The practical utility of CP was significantly enhanced by the development of split-conformal (or inductive) prediction, which separates the data into training and calibration sets, thereby amortizing the high computational cost of the original transductive methods (Papadopoulos et al., 2002; Lei et al., 2018). While the marginal coverage guarantee of $1 - \alpha$ is the central promise of CP, a primary focus of contemporary research is to improve the efficiency—that is, to produce the smallest possible prediction sets that still satisfy this guarantee.

Early CP methods often produced constant-width intervals, inefficient for heteroskedastic data. This spurred the development of adaptive methods to make prediction sets sensitive to local data characteristics. A widely adopted approach is Conformalized Quantile Regression (CQR) (Romano et al., 2019), which directly models the conditional quantiles of the target distribution, yielding intervals that naturally expand or contract with local uncertainty.

A parallel and highly influential line of work focuses on localizing the calibration procedure itself. Instead of treating the calibration set as a monolithic entity, these methods assign weights to calibration points based on their relevance to a new test point. Locally Weighted Conformal Prediction (Tibshirani et al., 2019) formalized this by using kernel functions to up-weight nearby calibration points, a concept further refined in numerous localized prediction techniques (Guan & Tibshirani, 2022). The principle of localization has also been applied to distribution shift, a common challenge in real-world deployments. Techniques for conformal prediction under covariate shift often involve weighting calibration scores by the ratio of training-to-test feature densities, effectively re-calibrating the model

to the target distribution (Tibshirani et al., 2019; Guan, 2020). We will explore this in greater detail when discussing the conditional coverage properties of AMDCP in Section 5, wherein we introduce G-AMDCP, a simple extension to AMDCP which leverages the principles of localized calibration to provide sharp group-level distribution-free conditional coverage guarantees.

**Improving Traditional CP:** Standard CP frameworks assume data exchangeability, which is violated by the temporal dependencies in data. To address this, methods like EnbPI (Ensemble-based Prediction Intervals) use a leave-one-out residual approach that is robust to non-stationarity (Xu & Xie, 2021). Other works, like SPCI (Sequential Predictive Conformal Inference) Xu & Xie (2023), adapt the CQR framework to online settings, using exponential weighting to prioritize recent calibration data. These advancements have made CP highly effective for unimodal regression and have been extended to structured data like time series (Xu & Xie, 2021). However, their core machinery is still predicated on a single, connected prediction region, which is insufficient for more complex distributional phenomena.

A major paradigm shift in the field has been the integration of powerful, flexible conditional density estimators from the deep learning toolkit. This allows the non-conformity score to be based on a point's estimated likelihood, a far more nuanced measure of atypicality than a simple residual. This move enables CP to tackle challenges like multimodality, which are common in real-world applications ranging from robotics (Choi et al., 2018) to computer vision (Prokudin et al., 2018) and forecasting.

**The "Black-Box" Conformalization Paradigm:** Early explorations into density-based CP utilized non-parametric methods like Kernel Density Estimators (KDEs) to define the non-conformity score, enabling the creation of arbitrarily shaped regions (Lei et al., 2011). However, the computational cost of KDEs and their poor scaling with dimensionality led to the dominance of the modern paradigm: treating powerful neural density models as "black-box" oracles. A non-conformity score is derived from the model's final, aggregate output—typically the total negative log-likelihood. This "black-box" strategy is employed by existing generative probabilistic methods, such as Distributional CP (DCP) (Chernozhukov et al., 2021), and the density-based mode in MAPIE (MAPIE developers, 2023). While other methods can also produce flexible sets such as CD-split (Izbicki et al., 2020), they often struggle to fully adapt to local manifold complexities, and the dominant strategy remains the use of neural density oracles.

This paradigm has been successfully applied to a wide array of generative architectures. For instance, normalizing flows have gained prominence, learning a complex, bijective transformation from the data distribution to a simple, tractable base distribution (e.g., a standard Gaussian) (Papamakarios et al., 2021). This has led to innovative CP methods like CONTRA, which constructs conformal sets in the simpler latent space of a flow model before mapping them back to the complex data space (Fang et al., 2025). However, a critical consequence of using the total log-likelihood is that the resulting prediction region is a single, connected super-level set of the estimated density. For multimodal distributions with well-separated modes, this forces the inclusion of the large, low-probability "valleys" between modes to maintain coverage, leading to overly conservative and potentially uninformative regions, as we visualize in Figure 1.

**The Frontier: Generating Non-Contiguous Prediction Regions:** Recognizing the fundamental limitations of connected sets for multimodal targets, a key frontier in CP research is the development of methods that can produce non-contiguous prediction regions. This is a challenging task, as the theoretical guarantees of CP must be maintained while allowing for complex, disjoint geometries.

The current frontier probabilistic method for non-contiguous regions, Probabilistic Conformal Prediction (PCP) (Wang et al., 2023) uses a generative model such as an MDN to produce a cloud of samples from the conditional distribution. A prediction region is then formed by taking the union of balls of a calibrated radius centered at each of these samples. This sampling-based construction is highly flexible and can capture arbitrary shapes. However, it comes with significant drawbacks: because it utilizes just the aggregate log-likelihood, the procedure is computationally intensive at inference time due to its reliance on generating many samples per test point, and the resulting region's geometry can be highly irregular and sensitive to the number of samples, potentially hindering interpretability.

**Our Contribution:** The existing literature presents a distinct trade-off: methods are either computationally efficient but restricted to connected regions (like CQR and DCP), or they are flexible enough

to be non-contiguous but are computationally burdensome and can produce irregular regions (like PCP). AMDCP bridges this gap by introducing a novel **"glass-box"** paradigm.

We now propose a "glass-box" or "component-aware" approach as one where the non-conformity score is designed to leverage the decomposed internal structure of a generative model. This contrasts with the standard aggregate approach (used by methods like DCP, PCP, or CONTRA) that uses only the final, summed log-likelihood, which we term as a "black-box" approach.

This decomposition is a unique property available to MDNs and is core to our "glass-box" approach and differs fundamentally from prior work. The inclusion of the data-dependent mixing weight is what allows us to "open the black box." While weights are mathematically present in the total log-likelihood score used by "black box" methods like DCP, the crucial information about which mode is most probable for a given point is effectively "averaged out" and lost. In contrast, our score directly leverages the mixing weights to actively shrink the prediction regions around the most probable mixture components—a selective, adaptive mechanism *impossible with black-box scores*. This direct inclusion enables our 'glass-box' innovation, an approach not readily available to monolithic density models like normalizing flows, and existing generative probabilistic models such as PCP, which rely on costly and inefficient sampling-based approaches to create non-contiguous regions. This "glass-box" decomposition also allows a constant-time analytic membership test, ensuring fast inference times. All of this is done while still retaining the architectural flexibility and robustness required by practitioners, as shown in our extensive robustness tests. And as our comprehensive evaluation demonstrates, AMDCP outperforms representative methods and is also 2-3x faster in training and faster at inference; and specifically in comparison to PCP, AMDCP produces predictions that are on 55% narrower on average while being an order of magnitude faster at inference time.

## 3 ADAPTIVE MIXTURE DENSITY FOR CONFORMAL PREDICTION (AMDCP)

In this section, we introduce the AMDCP framework. AMDCP combines the flexibility of MDNs to model complex conditional distributions $p(y|x)$. The key idea is to leverage the learned mixture structure to define a novel non-conformity score, enabling the construction of prediction regions that adapt their shape, orientation, and can capture multimodality based on local data characteristics. The general procedure for AMDCP is summarized in Algorithm 1.

Let $\{(X_i, Y_i)\}_{i=1}^N$ be a set of data points, where $X_i \in \mathcal{X}$ are features and $Y_i \in \mathcal{Y} \subseteq \mathbb{R}^m$ are the targets. We assume the pairs $(X_i, Y_i)$ are exchangeable. Given a new feature vector $X_{test}$, our goal is to construct a prediction region $\mathcal{C}_\alpha(X_{test})$ such that $\Pr(Y_{test} \in \mathcal{C}_\alpha(X_{test})) \geq 1 - \alpha$, where $1 - \alpha$ is the desired coverage level. We aim to achieve this while producing regions that are as narrow (efficient) as possible by adapting to local data characteristics like heteroskedasticity.

**Method overview.** AMDCP proceeds in three main steps:

1. **Mixture density network:** Model the conditional distribution $P(Y|X)$ using a mixture of Gaussians, where the mixture parameters (weights $\pi_j(X)$, means $\mu_j(X)$, and covariances $\Sigma_j(X)$) are outputs of a neural network conditioned on $X$.

2. **Adaptive non-conformity score:** Define a non-conformity score based on the Mahalanobis distance to the "best-fitting" mixture component, adjusted by mixture weight.

3. **Conformal calibration:** Apply standard split-conformal prediction using the scores from a calibration set to determine a threshold $\hat{q}_{1-\alpha}$. The resulting prediction regions can exhibit complex, adaptive geometries. For applications with potential distribution shifts (e.g., time series), this step can be augmented with weighted calibration. See Appendix A.1

### 3.1 MIXTURE DENSITY NETWORK

The foundation of AMDCP is an MDN that models $P(Y|X)$. For a given input $X$, the MDN outputs parameters for $K$ Gaussian components:

$$P(Y|X) = \sum_{j=1}^{K} \pi_j(X)\mathcal{N}(Y; \mu_j(X), \Sigma_j(X)),$$

(1)

---

**Algorithm 1** Adaptive Mixture Density for Conformal Prediction (AMDCP) - General procedure

---

**Require:** Training set $\mathcal{D}_{\text{train}}$, Calibration set $\mathcal{D}_{\text{cal}}$, new input $X_{\text{test}}$, target miscoverage rate $\alpha$.

    **Phase 1: Train MDN**

1: Train a Mixture Density Network (MDN) on $\mathcal{D}_{\text{train}}$ to model $P(Y|X)$ (Eq. 1), by optimizing parameters $\theta$ via Eq. 5.

    **Phase 2: Conformal Calibration**

2: **for** each $(X_i, Y_i) \in \mathcal{D}_{\text{cal}}$ **do**

3:    Compute MDN parameters $\{\pi_j(X_i; \theta), \mu_j(X_i; \theta), \Sigma_j(X_i; \theta)\}_{j=1}^K$.

4:    Calculate the non-conformity score $s_i = s(X_i, Y_i)$ as in Eq. 2.

5:    *(Optional for time series/drift)* Assign weight $w_i$ (e.g., $w_i = \exp[-\gamma(t_{\text{test}} - t_i)]$). If not using weighted calibration, set $w_i = 1$ for all $i$.

6: **end for**

7: Compute $\hat{q}_{1-\alpha} = \text{WeightedQuantile}(\{s_i, w_i\}_{i=1}^{M_{\text{cal}}}, 1 - \alpha)$. (If $w_i = 1$, this is standard quantile.)

    **Phase 3: Prediction Region**

8: For new input $X_{\text{test}}$, compute MDN parameters $\{\pi_j(X_{\text{test}}; \theta), \mu_j(X_{\text{test}}; \theta), \Sigma_j(X_{\text{test}}; \theta)\}_{j=1}^K$.

9: Form the prediction region $\mathcal{C}_\alpha(X_{\text{test}}) = \{Y : s(X_{\text{test}}, Y) \le \hat{q}_{1-\alpha}\}$ using Eq. 3.

10: **return** $\mathcal{C}_\alpha(X_{\text{test}})$

    *Note: For time series applications, an RMDN is used in Step 1, the time-series score (Eq. 6) in Step 4, and weighted calibration (assigning $w_i$ based on $t_i$) is typically used in Step 7.*

---

where $\sum_{j=1}^K \pi_j(X) = 1$ and $\pi_j(X) \ge 0$. The network is trained by maximizing the log-likelihood of the observed data on a training set $\mathcal{D}_{\text{train}}$.

While the covariance matrices used within the likelihood, $\Sigma_j(X)$, can be full covariance matrices (If full covariances are used, the predicted geometries would be general ellipsoids; the method supports both and the geometry adapts accordingly), diagonal covariances often suffice in practice and are computationally simpler, and are used in this paper.

### 3.2 ADAPTIVE NON-CONFORMITY SCORE

Given a data point $(X, Y)$ and the MDN parameters $\{\pi_j(X), \mu_j(X), \Sigma_j(X)\}_{j=1}^K$ learned from $\mathcal{D}_{\text{train}}$, we define the non-conformity score $s(X, Y)$ as:

$$s(X, Y) = \min_{1 \le j \le K} \left[ (Y - \mu_j(X))^\top \Sigma_j(X)^{-1} (Y - \mu_j(X)) + \log \det \Sigma_j(X) - 2 \log \pi_j(X) \right]. \quad (2)$$

This score constitutes a "glass-box" approximation of the negative log-likelihood[1]. The term $\log \det \Sigma_j(X)$ ensures that the score penalizes components with high uncertainty (large volume) even if they are close to the mean, preventing the selection of overly diffuse modes. The term $-2 \log \pi_j(X)$ favors components with higher mixture weights.

### 3.3 CONFORMAL CALIBRATION AND PREDICTION REGION CONSTRUCTION

We employ a standard split-conformal procedure. In addition to $\mathcal{D}_{\text{train}}$, we use a separate calibration set $\mathcal{D}_{\text{cal}} = \{(X_i, Y_i)\}_{i=1}^{M_{\text{cal}}}$, where points in $\mathcal{D}_{\text{cal}}$ are also exchangeable with $(X_{test}, Y_{test})$ and points in $\mathcal{D}_{\text{train}}$. First, for each $(X_i, Y_i) \in \mathcal{D}_{\text{cal}}$, we compute the non-conformity score $s_i = s(X_i, Y_i)$ using Equation 2, where the MDN parameters are fixed from training on $\mathcal{D}_{\text{train}}$. Then we determine the threshold $\hat{q}_{1-\alpha}$ as the $\lceil (M_{\text{cal}} + 1)(1 - \alpha) \rceil / M_{\text{cal}}$ empirical quantile of the scores $\{s_i\}_{i=1}^{M_{\text{cal}}}$. At inference, for a new input $X_{\text{test}}$, the prediction region is:

$$\mathcal{C}_\alpha(X_{\text{test}}) = \{Y \in \mathcal{Y} : s(X_{\text{test}}, Y) \le \hat{q}_{1-\alpha}\}. \quad (3)$$

---

[1]When using tied covariances $\Sigma_j(x) = \Sigma(x)$ for all $j$, the $\log \det \Sigma_j$ term becomes constant across components and is absorbed by the calibration threshold. In this case, the simplified score $\tilde{s}(x, y) = \min_j \{(y - \mu_j)^\top \Sigma^{-1}(y - \mu_j) - 2 \log \pi_j\}$ yields identical prediction sets. All theoretical guarantees, including Corollary 4.1.1, remain valid since level sets are preserved under constant shifts.

The region $\mathcal{C}_\alpha(X_{\text{test}})$ is effectively a finite union of up to $K$ (potentially overlapping) elliptical sets, each defined by one mixture component $j$ for which the term inside the $\min$ in Equation 2 is less than or equal to $\hat{q}_{1-\alpha}$.

Crucially, our framework retains modularity: the MDN serves as a lightweight output "head" that can be paired with any feature-extracting backbone, from simple MLPs to large Transformers. Our ablation studies in Appendix D confirm this flexibility, showing consistent performance across diverse architectures, extreme adversarial data distributions, and more. This makes AMDCP practical, allowing practitioners to leverage powerful pre-trained models to gain the benefits of our glass-box score without the cost of end-to-end training. Therefore, the MDN is not a limitation but an enabling component, providing the necessary probabilistic structure for our novel score while retaining the practical flexibility required for real-world applications.

## 4 THEORETICAL GUARANTEES FOR AMDCP

We establish two key theoretical guarantees for AMDCP. Detailed proofs and formal statements of regularity conditions are provided in Appendix B.

We analyze AMDCP in the split-conformal setting. Let $(X, Y) \in \mathcal{X} \times \mathbb{R}^d$ be a random pair with conditional density $p(y|x)$. The AMDCP framework utilizes a Mixture Density Network with $K$ components, outputting parameters $\theta(x) = \{\pi_j(x), \mu_j(x), \Sigma_j(x)\}_{j=1}^K$. Recall the AMDCP non-conformity score defined in Eq. 3. We define the population limit score $s^\star(x, y)$ based on the ideal population parameters $\theta^\star(x)$ as:

$$s^\star(x, y) := \min_{1 \le j \le K} \left\{ (y - \mu_j^\star(x))^\top \Sigma_j^\star(x)^{-1} (y - \mu_j^\star(x)) + \log \det \Sigma_j^\star(x) - 2 \log \pi_j^\star(x) \right\}. \quad (4)$$

This score is algebraically equivalent to $s^\star(x, y) = -2 \log(\max_j \pi_j^\star(x) \phi(y; \mu_j^\star, \Sigma_j^\star)) + C$, where $\phi$ is the Gaussian PDF and $C = d \log(2\pi)$. Let $\hat{q}_{1-\alpha}$ denote the empirical $(1 - \alpha)$ quantile of the scores computed on an independent calibration set of size $M$. The prediction region is $\hat{C}_\alpha(x) = \{y : s(x, y; \hat{\theta}_N) \le \hat{q}_{1-\alpha}\}$.

**Marginal Coverage Guarantee.** AMDCP satisfies the standard validity guarantee of split-conformal prediction due to the exchangeability of calibration and test scores. Following the established results in Vovk (2012) and Lei et al. (2018), for any finite training size $N$ and calibration size $M$, we have $\mathbb{P}(Y_{\text{test}} \in \hat{C}_\alpha(X_{\text{test}})) \ge 1 - \alpha$. As this result is well-known in the conformal literature, we focus the remainder of our analysis on the *efficiency* (volume) of the generated sets.

### 4.1 ASYMPTOTIC EFFICIENCY

To formalize efficiency, we rely on four regularity conditions (detailed in Appendix B). Briefly, **(A1) Parameter Consistency** assumes the learned parameters $\hat{\theta}_N$ converge to a limit $\theta^\star$; **(A2) Quantile Consistency** assumes the empirical calibration quantile converges to the true population quantile; and **(A3) Score Regularity** assumes the score is coercive (i.e., $s^\star(x, y) \to \infty$ as $\|y\| \to \infty$), which ensures prediction sets are bounded with finite volume. We also consider a fourth condition, **(A4) Asymptotic Mode Separation**, which assumes that in the limit, the overlap between mixture components becomes negligible at the decision boundary.

**Theorem 4.1** (Score-Relative Asymptotic Efficiency). *Under Assumptions (A1)–(A3), the volume of the AMDCP prediction region converges to the volume of the oracle region defined by the limit score* $s^\star$. *Let* $C_\alpha^{(s^\star)}(x) = \{y : s^\star(x, y) \le q_{1-\alpha}^\star\}$. *Then, as* $N, M \to \infty$:

$$\frac{\mathbb{E}[\text{Vol}(\hat{C}_\alpha(X))]}{\mathbb{E}[\text{Vol}(C_\alpha^{(s^\star)}(X))]} \xrightarrow{P} 1.$$

Theorem 4.1 establishes *internal consistency*: it guarantees that AMDCP yields the optimal set relative to its specific geometric construction (unions of ellipsoids). This result is crucial in proving that even if the "min" operator in Eq. 4 is an approximation of the true log-sum-exp likelihood, the method efficiently recovers the optimal set defined by that approximation.

**Corollary 4.1.1** (Oracle Highest-Density Efficiency)**.** *Let $C_\alpha^{HDR}(x)$ be the true Minimum-Volume Highest Density Region defined by the level sets of the true density $p(y|x)$. If Assumption (A4) holds, the level sets of the AMDCP score $s^\star$ converge to the level sets of $p(y|x)$. Consequently, the volume of the symmetric difference vanishes:*

$$\lambda(\hat{C}_\alpha(x)\Delta C_\alpha^{HDR}(x)) \xrightarrow{P} 0.$$

This corollary justifies the use of the computationally efficient "min" score. It demonstrates that our analytic union-of-ellipsoids construction is not merely a heuristic approximation; in the limit of well-separated modes, it converges exactly to the true minimum-volume sets derived from the full density, thereby incurring no asymptotic loss in statistical efficiency compared to "black-box" full-log-likelihood methods.

**Remark** (Generalization to Non-Gaussian Mixtures)**.** *While we implement AMDCP using Gaussian components to achieve analytic ellipsoidal sets, Assumption (A3) holds for any mixture of coercive densities (e.g., Laplace, Student-t). Theorem 4.1 implies that AMDCP would retain asymptotic efficiency with these alternative kernels, though the resulting geometry would change (e.g., unions of polytopes for Laplace). The Gaussian choice is motivated by the closed-form tractability of ellipsoids and the universal approximation capability of Gaussians (Bishop, 1994).*

## 5 EXPERIMENTAL RESULTS

We conducted extensive experiments to evaluate the performance of AMDCP compared to leading conformal prediction methods. Our experiments include both synthetic datasets and real-world datasets. Results are 5 trial averages unless otherwise stated. See Appendix C for data details. We conduct extensive ablation tests to demonstrate the robustness of AMDCP in challenging scenarios.

### 5.1 MAIN EXPERIMENTS

All experiments have a target coverage of 90% and rely on PyTorch and a three-layer MLP (ReLU activations, hidden dimension 64) for the core network, trained with the Adam optimizer (lr $10^{-3}$). All experiments are run on a MacBook Pro with an M1 Pro CPU. Performance is measured by the coverage rate (adherence to the target coverage) and the prediction region size (efficiency).

DCP (Chernozhukov et al., 2021) is a method utilizing MDNs, so we provide a direct comparison by using the exact same trained MDN (with $K = 5$) for both DCP and AMDCP. We do the same with PCP (Wang et al., 2023), as the current leading probabilistic method to generate non-contiguous prediction regions. This allows for a parsimonious comparison between the methods, comparing the union-of-ellipsoid method of AMDCP with the super-level-set of DCP, and the constant-time inference procedure of AMDCP compared to the sampling of PCP.

Other Baseline methods include (i) *Conformalized Quantile Regression (CQR)* (Romano et al., 2019), (ii) *Split Conformal Prediction (SCP)* with a Random Forest (Lei et al., 2018), (iii) *Locally Weighted Conformal Prediction (LWCP)* with Gradient Boosting (Guan, 2020), (iv) *Adaptive Conformal Prediction (ACP)* with a Random Forest, (Tibshirani et al., 2019), and (v) *Ensemble-based Prediction Intervals (EnbPI)* with Gradient Boosting (Xu & Xie, 2021). See Section 2 for a more detailed comparison of AMDCP with existing methods. For brevity, we include only the highest performing methods tested: PCP, DCP, CONTRA, and SPCI (for extended results, see Appendix C).

**Synthetic data evaluation.** Table 1 reports results on five synthetic datasets. AMDCP attains coverage close to the 90% target across all datasets (avg. 0.899). Restricting comparisons to methods with coverage $\geq 85\%$, AMDCP delivers the tightest or second–tightest intervals on 4/5 datasets—substantially so on *Mixture* ( 50% narrower than the next best, CONTRA) and clearly leading on *Multimodal* and *Regime-Switching* amongst those nearing target coverage. And while DCP can produce small intervals, it does so with vast undercoverage (avg. 0.799), underscoring that, despite sharing the same MDN, AMDCP's glass-box approach preserves target coverage while maintaining highly efficient predictions.

**Real-world data evaluation.** AMDCP's performance was evaluated on five diverse real-world forecasting tasks (Energy efficiency, Bike-sharing, Air-quality time-series, Protein property prediction

Table 1: Synthetic comparison (coverage = mean; width = mean $\pm 1$ SD). Best width in **bold**, second best underlined, considering only methods with coverage $\geq 0.89$ per dataset.

| Dataset | Metric | AMDCP | PCP | DCP | CONTRA | SPCI |
|---|---|---|---|---|---|---|
| Multimodal | Coverage | 0.901 | 0.985 | 0.784 | 0.909 | 0.898 |
| | Width | **2.088 ± 0.037** | 4.606 ± 0.140 | 1.851 ± 0.048 | 2.254 ± 0.045 | 2.637 ± 0.075 |
| Mixture | Coverage | 0.897 | 0.971 | 0.796 | 0.909 | 0.904 |
| | Width | **2.311 ± 0.096** | 8.167 ± 0.207 | 3.519 ± 0.227 | 4.640 ± 0.181 | 5.687 ± 0.342 |
| Skew-Het. | Coverage | 0.903 | 0.979 | 0.811 | 0.905 | 0.912 |
| | Width | 1.486 ± 0.044 | 2.780 ± 0.067 | 1.120 ± 0.035 | 1.437 ± 0.047 | **1.384 ± 0.041** |
| Regime-Switching | Coverage | 0.898 | 0.978 | 0.798 | 0.913 | 0.905 |
| | Width | **0.870 ± 0.034** | 1.846 ± 0.042 | 0.831 ± 0.018 | 1.128 ± 0.129 | 1.036 ± 0.029 |
| Multidimensional | Coverage | 0.898 | 0.980 | 0.806 | 0.907 | 0.890 |
| | Width | 1.296 ± 0.019 | 2.344 ± 0.046 | 1.090 ± 0.044 | 1.405 ± 0.056 | **1.267 ± 0.031** |

(Bio), and NYC Taxi location data), with results summarized in Table 2. For comparison of algorithms we provide the same MDN for AMDCP, PCP, and DCP. For brevity, we include only the highest performing methods tested (for extended results, see Appendix C). We compare against DCP, PCP, CONTRA, and SPCI, while also reporting training and inference times.

Table 2: Performance, train, and inference on datasets. For *width*, *train*, and *inference*, best in **bold**, second best underlined, considering only methods with coverage $\geq 0.89$ per dataset.

| Dataset | Metric | AMDCP | PCP | DCP | CONTRA | SPCI |
|---|---|---|---|---|---|---|
| Energy | Coverage | 0.903 | 0.992 | 0.786 | 0.891 | 0.844 |
| | Width (± SD) | **0.168 ± 0.081** | 1.032 ± 0.094 | 0.381 ± 0.078 | 0.356 ± 0.087 | 0.134 ± 0.004 |
| | Train (s) | **1.12** | **1.12** | 1.12 | 2.51 | 0.03 |
| | Infer (ms/pt) | **0.416** | 6.681 | 19.136 | 0.576 | 2.596 |
| Bike | Coverage | 0.921 | 0.990 | 0.785 | 0.900 | 0.893 |
| | Width (± SD) | **0.579 ± 0.449** | 1.418 ± 0.085 | 0.481 ± 0.049 | 0.617 ± 0.027 | 1.223 ± 0.035 |
| | Train (s) | 21.47 | 21.47 | 21.47 | 64.67 | **0.80** |
| | Infer (ms/pt) | **0.423** | 6.714 | 16.455 | 0.665 | 3.558 |
| Air | Coverage | 0.904 | 0.994 | 0.789 | 0.906 | 0.688 |
| | Width (± SD) | 0.941 ± 0.321 | 1.457 ± 0.078 | 0.573 ± 0.051 | **0.855 ± 0.052** | 0.501 ± 0.022 |
| | Train (s) | **0.79** | **0.79** | 0.79 | 2.56 | 0.06 |
| | Infer (ms/pt) | **0.416** | 6.708 | 19.136 | 0.577 | 2.645 |
| Bio | Coverage | 0.931 | 0.994 | 0.949 | 0.899 | 0.880 |
| | Width (± SD) | **2.125 ± 0.081** | 3.328 ± 0.105 | 2.561 ± 0.123 | 2.192 ± 0.083 | 14.368 ± 0.122 |
| | Train (s) | **11.24** | **11.24** | **11.24** | 31.23 | 1.55 |
| | Infer (ms/pt) | **0.395** | 6.715 | 15.998 | 0.571 | 3.400 |
| Taxi | Coverage | 0.902 | 0.996 | 0.950 | 0.901 | 0.900 |
| | Width (± SD) | **1.123 ± 0.040** | 3.318 ± 0.097 | 1.565 ± 0.019 | 1.356 ± 0.195 | 823.077 ± 5.761 |
| | Train (s) | 112.43 | 112.43 | 112.43 | 305.05 | **10.98** |
| | Infer (ms/pt) | **0.382** | 6.700 | 17.888 | 0.579 | 4.058 |

AMDCP's "glass-box" design provides a significant efficiency advantage over other methods that produce non-contiguous regions. PCP, the best probabilistic algorithm for comparison, forms prediction sets by expensive sampling methods; AMDCP instead produces an analytic union of ellipsoids from the MDN itself. The result retains PCP's ability to capture multimodality (non-contiguous sets), while also providing sharper, interpretable geometries and the observed speed/efficiency gains. We note that AMDCP similarly outperforms DCP, CONTRA and SPCI.

AMDCP's inference time is consistently an **order of magnitude faster than PCP (e.g., 17x faster on the Bio dataset), while producing intervals that are on average 55% narrower**. Furthermore, its training time **is often 2-3x faster than heavier models like CONTRA**, making it a highly practical choice for achieving flexible, sharp uncertainty estimates without prohibitive computational cost. These results speak to AMDCP's significant potential for practical, reliable uncertainty quantification in diverse real-world systems when the underlying data distributions are complex and might not be ideally suited to simpler quantile or unimodal assumptions, and where maintaining valid coverage is paramount in real-time situations where inference time and train time are critical.

## 5.2 ABLATION STUDIES AND ROBUSTNESS ANALYSIS

Full details are provided in Appendix D.

**Robustness to Architecture choice.** AMDCP's performance is agnostic to neural backbone architecture. We evaluated it with MLP, Transformer, LSTM, and ResNet backbones on various synthetic datasets (see Appendix D.2). Across all architectures, AMDCP consistently achieved empirical coverage near the 90% target (typically 88-91%) with prediction region widths differing by less than 5% on the same dataset. This flexibility allows AMDCP to be integrated as a lightweight "head" on top of powerful pre-trained feature extractors, retaining high performance while being significantly faster to train than current leading methods like CONTRA which require end-to-end training of more complex models. This is also important as this means that the order of magnitude faster inference will hold across model architectures, enabling AMDCP to be used in a wide variety of specialized domains with specialized backbone architectures.

**Robustness to Model Misspecification ($K$).** A critical hyperparameter for mixture models is the number of components $K$. We performed an extensive analysis and found AMDCP to be highly robust to the choice of $K$, mitigating concerns about extensive tuning. Our results, detailed in Appendix D.1, show that performance on standard multimodal data stabilizes quickly once $K$ is sufficient to capture the primary data modes. More importantly, we conducted stress tests demonstrating that AMDCP is remarkably stable against severe *over-specification* on simple data (gracefully ignoring extraneous components, with a mere 9% increase in interval width while maintaining target coverage even when $K$ was misspecified by two orders of magnitude ($K = 100$), see Table 7) and degrades slowly and predictably when severely *under-specified* on exceptionally complex data, where it still produces more efficient intervals than existing alternatives. This robustness makes AMDCP a practical choice for real-world applications where true data complexity is unknown.

**Robustness to Under- and Over-Fitting.** We tested scenarios where the MDN was intentionally underfit or severely overfit. Even in these extreme cases, coverage remained within 1% of the well-trained model, with only a 10-12% increase in interval width. This confirms that the benefits of AMDCP do not disappear when the MDN underperforms. See Table 11 in Appendix D for details.

Table 3: Performance of Existing Probabilistic Methods with CQR Baseline under Distributional Misspecification (Target Coverage: 90%). Results are averaged over 5 runs ($\mu \pm \sigma$). Best width in **bold** considering only methods with coverage $\geq 0.89$ per dataset.

| (a) Asymmetric Skewed Data | | | | (b) Symmetric Heavy-Tailed Data | | |
|---|---|---|---|---|---|---|
| **Method** | **Cov.** | **Width** | | **Method** | **Cov.** | **Width** |
| AMDCP | $0.899 \pm 0.006$ | $\mathbf{0.987 \pm 0.032}$ | | AMDCP | $0.900 \pm 0.001$ | $\mathbf{4.268 \pm 0.001}$ |
| CQR | $0.895 \pm 0.017$ | $1.016 \pm 0.034$ | | CQR | $0.898 \pm 0.012$ | $4.389 \pm 0.254$ |
| DCP | $0.877 \pm 0.000$ | $0.942 \pm 0.000$ | | DCP | $0.899 \pm 0.000$ | $4.365 \pm 0.000$ |
| PCP | $0.995 \pm 0.002$ | $2.038 \pm 0.154$ | | PCP | $0.994 \pm 0.003$ | $15.154 \pm 1.347$ |

**Robustness to Extreme Distributions.** We further stress-tested AMDCP on data with heavy-tailed (t-distribution) and skewed bimodal noise. In these challenging scenarios designed to violate the MDN's mixture-of-Gaussian assumptions, AMDCP successfully maintained target coverage and produced intervals that were 3-5% narrower than CQR. While DCP achieves competitive interval widths and even narrower intervals on skewed data, it drastically falls short in coverage (87.7% vs. 90% target) and produces slightly wider intervals on heavy-tailed data (2.3% wider). AMDCP intervals were over 50-70% narrower than PCP, demonstrating the practical robustness of our glass-box method. This performance under assumption violations demonstrates that AMDCP's distributional framework provides robust inductive bias rather than restrictive constraints on applicability. The consistent efficiency gains across data structures suggest that the method's theoretical assumptions translate to practical benefits even when not perfectly satisfied. See Table 3 and Appendix D for details.

**Robustness to Extreme Distributions and Higher Dimensions.** We further stress-tested AMDCP on data with heavy-tailed ($t$-distribution) and skewed bimodal noise. In these challenging scenarios designed to violate the MDN's mixture-of-Gaussian assumptions, AMDCP successfully maintained target coverage and produced intervals that were 3–5% narrower than CQR. While DCP achieves competitive interval widths and even narrower intervals on skewed data, it drastically falls short in

coverage (87.7% vs. 90% target) and produces slightly wider intervals on heavy-tailed data (2.3% wider). AMDCP intervals were over 50-70% narrower than PCP.

We also evaluated scalability to high-dimensional outputs ($d_Y = 2, 5, 10$) using complex mixture data. While standard methods often suffer from volume explosion in high dimensions, AMDCP's analytic score scales efficiently: in 10-dimensional synthetic tests, AMDCP achieved an average width of 10.61, significantly outperforming CQR (11.96) and PCP (23.89). We validated this on real-world 5-dimensional Air Quality data, where AMDCP reduced prediction volume by 23% compared to CQR and by 50% compared to generative baselines (PCP/DCP), while maintaining valid coverage. On 2-dimensional Indoor Localization data, AMDCP matched the efficiency of non-parametric CQR while remaining $\sim$40% narrower than existing generative baselines. This results in conjunction with the synthetic multidimensional dataset in Section 4, demonstrate that AMDCP provides robust performance in multidimensional settings. See Appendix D.6 for full results and further experiments.

**Conditional Coverage Analysis.** We analyze how empirical coverage varies across different regions of the feature space as a practical measure of conditional reliability. Our analysis in Appendix E, employs both binned conditional coverage and coverage within random local Euclidean balls (COV-EUCS). AMDCP improved worst-bin coverage by up to 35 percentage points compared to SCP, while also frequently achieving higher worst-ball coverage than SCP. This improvement is likely due to AMDCP's ability to adapt its prediction regions to data complexities via its per-component non-conformity score. However, the degree of conditional validity still partly depends on how well the underlying MDN captures the true conditional distribution. Beyond these empirical checks, we provide an *approximate* group-conditional coverage result for AMDCP by adapting the conditional-coverage arguments for MDNs from DCP (Chernozhukov et al., 2021). To obtain *sharp* finite-sample, distribution-free guarantees, we extend AMDCP to **G-AMDCP**, a split-conformal variant that calibrates *separately within known discrete subgroups*. Formal definitions, proofs, and implementation details (with negligible overhead relative to standard AMDCP) are given in Appendix E.3.

# 6 DISCUSSION

To maximize utility, practitioners should distinguish when AMDCP is necessary (See Appendix F for further discussion). AMDCP is best suited for scenarios involving multimodality, heteroskedasticity, extreme data distributions, situations where fine-tuning is difficult, or where non-contiguous regions are required. Conversely, for datasets that are strongly unimodal and homoskedastic, our results show that simpler methods may suffice and perform comparably to AMDCP. In low-data scenarios, AMDCP's performance can degrade as the MDN may struggle to learn the conditional density, and simpler non-parametric methods may be more stable. Our analytic score also relies on the "max-component" approximation. While Corollary 4.1.1 proves asymptotic convergence to optimality, in regimes with heavily overlapping modes, the "min" operator produces sharp geometric cusps at the intersection of ellipsoids, whereas the full likelihood density creates smooth valleys, representing the trade-off for achieving $O(1)$ inference speed. Finally, training an MDN imposes incremental costs (albeit less than leading methods) that can be avoided by using a simpler non-parametric method.

In addition to these weaknesses, we also see concrete avenues for future improvement. Our guarantees are finite-sample *marginal*—full distribution-free conditional coverage is unattainable in general—so practitioners should not interpret our sets as instance-wise probabilities. When discrete subgroups are available, we provide a simple extension, G-AMDCP, that yields sharp, distribution-free, finite-sample coverage conditional on group membership (App. E.3). AMDCP also fixes the number of mixture components $K$; although ablations indicate robustness across a broad range (Appendix D.1) of $K$, developing data-adaptive mechanisms warrants further study. Finally, as with any conformal method, distribution shift can degrade calibration; periodic recalibration mitigates but may not eliminate this risk, underscoring the need for monitoring failure modes in production.

Across synthetic and real-world datasets, AMDCP yields sharper prediction sets at comparable coverage to strong baselines and an order-of-magnitude faster inference than current sampling-based probabilistic methods. By designing a "glass-box" score that is synergistic with the internal architecture of the MDN, AMDCP offers a novel, practical, and efficient solution for rapidly generating adaptive prediction regions for complex data. We view AMDCP as a practical, glass-box approach to non-contiguous conformal sets: it preserves the rigor of CP, aligns set geometry with local uncertainty through a per-component score, and remains computationally light to enable real-world deployment.

**ICLR Reproducibility Statement.** We provide exact descriptions of the algorithm in Section 3 and Appendix 1. Section 5 contains detailed explanations of the hardware and software used in experiments, as well as any and all hyperparameters. Appendix C contains details on both the synthetic and real-world datasets used in experiments. We fix random seeds for Python/NumPy/PyTorch in all scripts. Each result is averaged over 5 independent seeds; we report mean and standard deviation. Lastly, we will publicly release an anonymized repository containing an implementation of AMDCP.

**ICLR Ethics Statement.** We affirm adherence to the ICLR Code of Ethics for submission, review, and discussion. Our experiments use publicly available, de-identified datasets (e.g., UCI repositories and NYC TLC trip data) under their stated licenses; we do not collect human subjects data and therefore required no IRB approval. To reduce risk of harm and misuse, we emphasize that conformal prediction provides marginal (and group-conditional where explicitly calibrated) coverage guarantees and should not be interpreted as individualized probabilities in high-stakes settings; we report conditional-coverage diagnostics and provide a simple group-wise calibration variant (G-AMDCP), but practitioners should conduct domain-specific fairness and shift-robustness audits before deployment. An LLM was used to identify and polish grammatical errors in writing, with author oversight. We reaffirm that we have read and will adhere to the ICLR Code of Ethics across all conference activities, including submission, reviewing, and discussion. We do not foresee dual-use risks beyond those common to predictive modeling, but we strongly caution against unvetted use in safety-critical applications without additional governance. All potential conflicts of interest and funding sources, if any, will be disclosed transparently in the camera-ready version.

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

# A ALGORITHM EXTENSIONS

The network is trained by maximizing the log-likelihood of the observed data on a training set $\mathcal{D}_{\text{train}}$:

$$\mathcal{L} = \sum_{(X_i, Y_i) \in \mathcal{D}_{\text{train}}} \log \left( \sum_{j=1}^{K} \pi_j(X_i) \mathcal{N}(Y_i; \mu_j(X_i), \Sigma_j(X_i)) \right). \tag{5}$$

## A.1 EXTENSIONS FOR TIME SERIES FORECASTING.

Below we present some connections to existing work to inspire some modifications for time series data. However, we note that AMDCP performs well as-is in time series data, as evidenced in our experimental results in Section 5. Hence, the analysis below is purely a note with regards to related work and is beyond the scope of this paper.

**Recurrent mixture density networks (RMDN).** For time series, the MDN in Section 3.1 can be implemented using recurrent neural layers (e.g., LSTM, GRU) to effectively capture temporal dependencies from the history $X_{1:t}$. The RMDN models this sequence by predicting parameters for a mixture of $K$ Gaussian trajectories, where each trajectory $j$ has means $\{\mu_{j,k}(X_{1:t})\}_{k=1}^{h}$ and covariances $\{\Sigma_{j,k}(X_{1:t})\}_{k=1}^{h}$ for each step $k = 1, \ldots, h$, and an overall mixing weight $\pi_j(X_{1:t})$.

**Time-Series non-conformity score.** The general non-conformity score (Equation 2) adapts naturally. The score sums contributions over the forecast horizon $h$:

$$s(X_{1:t}, Y_{t+1:t+h}) = \min_{1 \le j \le K} \left( \left[ \sum_{k=1}^{h} \beta_k \left( (Y_{t+k} - \mu_{j,k})^\top \Sigma_{j,k}^{-1} (Y_{t+k} - \mu_{j,k}) + \log \det \Sigma_{j,k} \right) \right] \right. \\ \left. - 2 \log(\pi_j(X_{1:t})) \right). \tag{6}$$

where $\beta_k > 0$ are optional weights. The minimization is over a common component index $j$ across all steps, inducing a single coherent forecast trajectory through the mixture components.

**Weighted conformal calibration for temporal shift.** Time series data often violate the exchangeability assumption due to temporal drift, where the data-generating distribution $P_t(X, Y)$ changes over time $t$. To mitigate this, the calibration step (Section 3.3) can be modified using weighted conformal calibration (Tibshirani et al., 2019). This prioritizes more recent, and presumably more relevant, calibration data. Our choice of exponentially decaying weights $w_k = \exp[-\gamma(t_{\text{test}} - t_k)]$ is designed to make $\Delta$ small. By assigning large normalized weights $\nu_k$ only to recent calibration points (small $t_{\text{test}} - t_k$), for which we expect the distributional drift $d_{TV}(G_k, G_{\text{test}})$ to be small (assuming the data-generating process changes gradually over time), the sum $\Delta$ can be kept small. This provides a theoretical basis for why weighted conformal prediction can achieve coverage close to the target $1 - \alpha$ even when the strict exchangeability assumption is violated by temporal drift. See Tibshirani et al. (2019) for the basis of this approach.

## B  PROOFS OF THEORETICAL GUARANTEES

### B.1  SETUP AND DEFINITIONS

We consider the split-conformal prediction setting. Let $(X, Y) \in \mathcal{X} \times \mathbb{R}^d$ be a random vector. We are given a training set $\mathcal{D}_{\text{train}}$ of size $N$, a calibration set $\mathcal{D}_{\text{cal}}$ of size $M$, and a test point $(X_{\text{test}}, Y_{\text{test}})$. We assume exchangeability of $\mathcal{D}_{\text{cal}} \cup \{(X_{\text{test}}, Y_{\text{test}})\}$.

The AMDCP framework estimates $p(y|x)$ using a Mixture Density Network with $K$ Gaussian components, outputting parameters $\hat{\theta}_N(x) = \{\hat{\pi}_j(x), \hat{\mu}_j(x), \hat{\Sigma}_j(x)\}_{j=1}^K$. The non-conformity score is:

$$s(x, y; \theta) = \min_{1 \leq j \leq K} \left\{ (y - \mu_j)^\top \Sigma_j^{-1} (y - \mu_j) + \log \det \Sigma_j - 2 \log \pi_j \right\}. \tag{7}$$

Let $s^\star(x, y) = s(x, y; \theta^\star)$ denote the population limit score, where $\theta^\star(x)$ denotes the probability limit of $\hat{\theta}_N(x)$ (i.e., the parameters to which the MDN converges under maximum likelihood training).

**Remark** (Score Equivalence). *Define $Q_j(y) = (y - \mu_j)^\top \Sigma_j^{-1} (y - \mu_j) + \log \det \Sigma_j - 2 \log \pi_j$. Using the Gaussian PDF $\phi(y; \mu, \Sigma) = (2\pi)^{-d/2} |\Sigma|^{-1/2} \exp(-\frac{1}{2}(y - \mu)^\top \Sigma^{-1}(y - \mu))$, we have $Q_j(y) = -2\log(\pi_j \phi_j(y)) - d\log(2\pi)$. Thus $s^\star(x, y) = -2\log(\max_j \pi_j^\star \phi_j(y)) - d\log(2\pi)$, showing our score equals the negative log of the dominant mixture component (up to a universal constant).*

### B.2  REGULARITY ASSUMPTIONS

(A1) **Parameter Consistency.** $\hat{\theta}_N(x) \xrightarrow{P} \theta^\star(x)$ as $N \to \infty$, where $\Sigma_j^\star(x) \succ 0$ and $\pi_j^\star(x) > 0$ for all $j$. Moreover, there exist constants $0 < \underline{\lambda} < \bar{\lambda} < \infty$ and $\underline{\pi} > 0$ such that $\underline{\lambda} I \preceq \Sigma_j^\star(x) \preceq \bar{\lambda} I$ and $\pi_j^\star(x) \geq \underline{\pi}$ uniformly over $x \in \mathcal{X}$ and $j \in [K]$.

(A2) **Quantile Regularity.** The CDF $F_{S^\star}$ of $S^\star = s^\star(X, Y)$ is continuous and strictly increasing in a neighborhood of $q_{1-\alpha}^\star := F_{S^\star}^{-1}(1 - \alpha)$. By the Glivenko-Cantelli theorem, the empirical quantile satisfies $\hat{q}_{1-\alpha} \xrightarrow{P} q_{1-\alpha}^\star$ as $M \to \infty$.

(A3) **Score Coercivity.** For almost every $x$, $s^\star(x, y) \to \infty$ as $\|y\| \to \infty$. Furthermore, for any $c \in \mathbb{R}$, the level set boundary $\partial \{y : s^\star(x, y) = c\}$ has Lebesgue measure zero.

(A4) **Asymptotic Mode Separation.** Let $p(y|x) = \sum_j \pi_j^\star \phi_j(y)$ be the limiting mixture density. As a separation parameter $\delta \to \infty$ (encoding inter-mode distance), the log-sum-exp gap satisfies: $\sup_y \left| \log(\sum_j \pi_j^\star \phi_j) - \log(\max_j \pi_j^\star \phi_j) \right| \to 0$.

### B.3  PROOF OF THEOREM 4.1 (SCORE-RELATIVE ASYMPTOTIC EFFICIENCY)

We start with 3 helpful lemmas in applying the Dominated Convergence Theorem for predicted region volumes:

1.

  **Lemma B.1** (Pointwise Score Convergence). *Under (A1), for any fixed $(x, y)$: $s(x, y; \hat{\theta}_N) \xrightarrow{P} s^\star(x, y)$.*

  *Proof.* The score $s(x, y; \theta) = \min_j Q_j(y; \theta)$ where $Q_j$ involves matrix inversion, log-determinant, and logarithm of mixture weights. Each $Q_j$ is continuous in $\theta$ on the domain $\{\theta : \Sigma_j \succ 0, \pi_j > 0\}$. Since $\theta^\star$ lies in the interior of this domain by (A1), and $\hat{\theta}_N \xrightarrow{P} \theta^\star$, the Continuous Mapping Theorem yields $Q_j(y; \hat{\theta}_N) \xrightarrow{P} Q_j(y; \theta^\star)$ for each $j$. Since $\min$ over finitely many convergent sequences converges to the minimum of the limits, $s(x, y; \hat{\theta}_N) \xrightarrow{P} s^\star(x, y)$. $\square$

2.

  **Lemma B.2** (Indicator Convergence). *Under (A1)–(A3), for Lebesgue-almost every $y \in \mathbb{R}^d$: $\mathbb{I}(s(x, y; \hat{\theta}_N) \leq \hat{q}_{1-\alpha}) \xrightarrow{P} \mathbb{I}(s^\star(x, y) \leq q_{1-\alpha}^\star)$.*

*Proof.* Define $A_N = s(x, y; \hat{\theta}_N) - \hat{q}_{1-\alpha}$ and $A = s^\star(x, y) - q^\star_{1-\alpha}$. By Lemma B.1 and (A2), $(s(x, y; \hat{\theta}_N), \hat{q}_{1-\alpha}) \xrightarrow{P} (s^\star(x, y), q^\star_{1-\alpha})$, hence $A_N \xrightarrow{P} A$. The function $h(z) = \mathbb{I}(z \leq 0)$ is continuous except at $z = 0$. By the Continuous Mapping Theorem, $h(A_N) \xrightarrow{P} h(A)$ provided $\mathbb{P}(A = 0) = 0$, i.e., $y \notin \partial\{z : s^\star(x, z) = q^\star_{1-\alpha}\}$. By (A3), this boundary has Lebesgue measure zero. $\qquad\square$

3.

**Lemma B.3** (Uniform Boundedness of Prediction Sets). *Under (A1) and (A3), there exists $R < \infty$ such that $\mathbb{P}(\hat{C}_\alpha(x) \subseteq B(0, R)) \to 1$ as $N, M \to \infty$.*

*Proof.* By the uniform eigenvalue bounds in (A1), for any $\theta$ in a neighborhood of $\theta^\star$:

$$Q_j(y; \theta) \geq \underline{\lambda}^{-1} \|y - \mu_j\|^2 - d \log \bar{\lambda} - 2 \log(1/\underline{\pi}).$$

Thus $s(x, y; \theta) \geq \underline{\lambda}^{-1}(\|y\| - \max_j \|\mu_j\|)^2 - C_0$ for a constant $C_0$. Since $\hat{\theta}_N \xrightarrow{P} \theta^\star$ and $\hat{q}_{1-\alpha} \xrightarrow{P} q^\star_{1-\alpha}$, for any $\epsilon > 0$, with probability $\to 1$: $\hat{q}_{1-\alpha} < q^\star_{1-\alpha} + \epsilon$ and $\hat{\theta}_N$ lies in the neighborhood where the bound holds. Choose $R$ such that $\underline{\lambda}^{-1}(R - \sup_\theta \max_j \|\mu_j\|)^2 - C_0 > q^\star_{1-\alpha} + \epsilon$. Then $\|y\| > R \Rightarrow s(x, y; \hat{\theta}_N) > \hat{q}_{1-\alpha}$, so $y \notin \hat{C}_\alpha(x)$. $\qquad\square$

***Proof of Theorem 4.1.*** Let $V_N = \int_{\mathbb{R}^d} \mathbb{I}(s(x, y; \hat{\theta}_N) \leq \hat{q}_{1-\alpha})\, dy$ and $V^\star = \int_{\mathbb{R}^d} \mathbb{I}(s^\star(x, y) \leq q^\star_{1-\alpha})\, dy$.

By Lemma B.3, with probability $\to 1$, both integrands are supported on $B(0, R)$. Thus we may write $V_N = \int_{B(0,R)} \mathbb{I}(\cdot)\, dy$ with high probability. By Lemma B.2, the integrand converges in probability for a.e. $y$. Since $\mathbb{I}(\cdot) \leq \mathbb{I}(y \in B(0, R))$ and $\lambda(B(0, R)) < \infty$, the Dominated Convergence Theorem (extended to convergence in probability) yields $V_N \xrightarrow{P} V^\star$.

For the ratio of expectations: by (A2), $V^\star > 0$ (since strict monotonicity implies the set has positive measure). By uniform integrability (bounded by $\lambda(B(0, R))$), $\mathbb{E}[V_N] \to \mathbb{E}[V^\star]$, hence

$$\mathbb{E}[V_N]/\mathbb{E}[V^\star] \to 1$$

$\qquad\square$

## B.4 PROOF OF COROLLARY 4.1.1 (CONVERGENCE TO HIGHEST DENSITY REGION)

*Proof.* Let $s_{\text{HDR}}(y) = -2 \log p(y|x)$ where $p(y|x) = \sum_j \pi^\star_j \phi_j(y)$. Define $C^{\text{HDR}}_\alpha = \{y : s_{\text{HDR}}(y) \leq q_{\text{HDR}}\}$ and $C^\star_\alpha = \{y : s^\star(y) \leq q^\star\}$, where thresholds achieve $(1 - \alpha)$ coverage.

**Step 1 (Score Inequality).** Using $\max_j a_j \leq \sum_j a_j \leq K \max_j a_j$ for $a_j = \pi^\star_j \phi_j(y) > 0$:

$$s^\star(y) \leq s_{\text{HDR}}(y) \leq s^\star(y) + 2 \log K.$$

Define $\epsilon := \sup_y |s^\star(y) - s_{\text{HDR}}(y)|$. The bound $\epsilon \leq 2 \log K$ holds globally. Under (A4), $\epsilon \to 0$ as modes separate.

**Step 2 (Quantile Convergence).** Since $|s^\star(Y) - s_{\text{HDR}}(Y)| \leq \epsilon$ almost surely, the CDFs satisfy $F_{s^\star}(t) \leq F_{s_{\text{HDR}}}(t + \epsilon)$ and $F_{s_{\text{HDR}}}(t) \leq F_{s^\star}(t + \epsilon)$. For the $(1 - \alpha)$-quantiles: $|q^\star - q_{\text{HDR}}| \leq \epsilon + o(1)$ by continuity of $F_{s^\star}^{-1}$ (from (A2)).

**Step 3 (Symmetric Difference).** For $y \in C^\star_\alpha \Delta C^{\text{HDR}}_\alpha$, either (i) $s^\star(y) \leq q^\star$ but $s_{\text{HDR}}(y) > q_{\text{HDR}}$, or (ii) vice versa. In case (i): $s_{\text{HDR}}(y) \leq s^\star(y) + \epsilon \leq q^\star + \epsilon$ and $s_{\text{HDR}}(y) > q_{\text{HDR}} \geq q^\star - \epsilon - o(1)$, so $|s_{\text{HDR}}(y) - q_{\text{HDR}}| \leq 2\epsilon + o(1)$. Thus:

$$C^\star_\alpha \Delta C^{\text{HDR}}_\alpha \subseteq \{y : |s^\star(y) - q^\star| \leq 2\epsilon + o(1)\}.$$

By (A3), the boundary $\{y : s^\star(y) = q^\star\}$ has measure zero. By continuity of $s^\star$ and finiteness of level sets (from coercivity), the $\delta$-neighborhood of this boundary has measure $O(\delta)$ as $\delta \to 0$. Hence $\lambda(C^\star_\alpha \Delta C^{\text{HDR}}_\alpha) \to 0$ as $\epsilon \to 0$. $\qquad\square$

# C  Main experiments details

## C.1  Experimental Protocol and Metrics

To ensure robust and reproducible comparisons, we adhere to the following protocol for all experiments reported in Tables 1, 2, and 3:

**Repetitions and Splitting.** We report results averaged over 5 independent trials. For **Synthetic Data**, each trial involves regenerating the entire dataset from the data-generating process, followed by a random 60/20/20 split for training, calibration, and testing. For **Real-World Data**, each trial involves a random reshuffling of the full dataset followed by a fresh 60/20/20 split. In all cases, the MDN and baseline models are re-initialized and trained from scratch for each trial.

**Evaluation Metrics.** Reported values denote the Mean $\pm$ 1 Standard Deviation (SD) across the 5 trials. **Coverage:** Calculated as the fraction of points in the **entire test set** ($N_{test}$) that fall within the predicted region. **Width:** Calculated as the average volume (or length) of the prediction regions across the **entire test set**.

We refer to Appendix D.3 for ablations on dataset size.

## C.2  Synthetic data experiments

We consider five synthetic data-generating processes, designed to simulate real-world data challenges:

*(1) Multimodal data.* We define

$$
Y \mid X \sim \begin{cases}
\mathcal{N}\big(\sin(2X),\, 0.1 + 0.2|X|\big) & \text{with probability } 0.4, \\
\mathcal{N}\big(0.5X + 0.5\cos(3X),\, 0.2 + 0.1|\cos(X)|\big) & \text{with probability } 0.3, \\
\mathcal{N}\big(0.3X^2 - 0.5,\, 0.15 + 0.15|X|/3\big) & \text{with probability } 0.3.
\end{cases}
$$

*(2) Mixture data.* We sample from a mixture of Gaussians with input-dependent weights, means, and variances:

$$
\begin{aligned}
\pi_1(X) &= 0.5 + 0.4\sin(X), \quad \pi_2(X) = 1 - \pi_1(X), \\
\mu_1(X) &= X + 1, \quad \mu_2(X) = -X - 0.5, \\
\sigma_1(X) &= 0.2 + 0.1|X|, \quad \sigma_2(X) = 0.3 + 0.05X^2.
\end{aligned}
\tag{8}
$$

*(3) Skewed heteroskedastic data.* We generate a cubic mean function with heteroskedastic and skewed noise:

$$
Y \mid X \sim \mathcal{N}\big(0.1X^3 + 0.5X,\, 0.2 + 0.15|X|\big),
$$

*(4) Regime-switching data.* Inputs are drawn uniformly in $[-3, 3]$ and split into three regimes: (a) $x < -1$ with cubic patterns and varying noise, (b) $-1 \le x < 1$ with bimodal behavior, and (c) $x \ge 1$ with exponential growth and skew. Noise scales with $|x|$ differently in each regime.

*(5) Multidimensional data (2D input).* Here, $X = (x_1, x_2)$ in $[-3, 3]^2$, with a base mean $0.5\,x_1 + 0.3\,x_2 + 0.4\,x_1 x_2 + 0.2\,\sin(x_1 x_2)$ and noise variance $0.1 + 0.1\,\sqrt{x_1^2 + x_2^2}$.

### C.2.1  Extended Results

Table 4 shows that AMDCP delivers the tightest widths on 4/5 synthetic datasets (Multimodal, Mixture, Regime-Switching, Multidimensional) while maintaining coverage $\ge 0.85$. On Skew-Het, LWCP is narrowest (coverage 0.866), with CQR second; AMDCP is close behind with higher coverage (0.903). Across datasets, CQR is most often the second-best eligible method, whereas SCP/ACP/EnbPI typically yield wider intervals. Overall, AMDCP remains the leading baseline under the $\ge 0.85$ coverage filter.

## C.3  Real-world data

For our real-world evaluations, we utilized several publicly available datasets commonly used in forecasting and regression tasks. All datasets were preprocessed by splitting into training, calibration,

Table 4: Synthetic comparison (coverage = mean; width = mean $\pm 1$ SD) for remaining methods. Best width in **bold**, second best underlined, considering only methods with coverage $\geq 0.89$ per dataset.

| Dataset | Metric | AMDCP | CQR | SCP | LWCP | ACP | EnbPI |
|---|---|---|---|---|---|---|---|
| Multimodal | Coverage | 0.901 | 0.887 | 0.903 | 0.855 | 0.891 | 0.897 |
| | Width | **2.088 ± 0.037** | 2.173 ± 0.027 | 3.375 ± 0.168 | 2.242 ± 0.077 | 3.384 ± 0.879 | 2.685 ± 0.131 |
| Mixture | Coverage | 0.897 | 0.894 | 0.894 | 0.878 | 0.862 | 0.890 |
| | Width | **2.311 ± 0.096** | 4.065 ± 0.077 | 6.701 ± 0.318 | 4.765 ± 0.335 | 6.026 ± 2.018 | 5.338 ± 0.234 |
| Skew-Het. | Coverage | 0.903 | 0.909 | 0.901 | 0.866 | 0.892 | 0.904 |
| | Width | 1.486 ± 0.044 | **1.442 ± 0.052** | 1.805 ± 0.029 | 1.342 ± 0.047 | 1.885 ± 0.484 | 1.517 ± 0.082 |
| Regime-Switching | Coverage | 0.898 | 0.892 | 0.892 | 0.863 | 0.898 | 0.896 |
| | Width | **0.870 ± 0.034** | 0.993 ± 0.022 | 1.535 ± 0.092 | 1.046 ± 0.032 | 1.594 ± 0.199 | 1.406 ± 0.042 |
| Multidimensional | Coverage | 0.898 | 0.905 | 0.904 | 0.816 | 0.905 | 0.899 |
| | Width | **1.296 ± 0.019** | 1.339 ± 0.024 | 1.636 ± 0.070 | 1.296 ± 0.032 | 1.763 ± 0.492 | 1.669 ± 0.062 |

and test sets (with a 60%-20%-20% ratio respectively, using a fixed random seed for each run for reproducibility), followed by standardization of features and targets based on the training set statistics. For time-series datasets, features often included lagged values of the target variable and time-based features (e.g., hour of day, day of week).

- **Energy efficiency data** ($N = 768$; $d_X = 8, d_Y = 1$). The data was obtained from the UCI Machine Learning Repository (Tsanas & Xifara, 2012).This was a regression task to predict the Heating Load (Y1) of residential buildings. The 8 input features are: Relative Compactness (X1), Surface Area (X2), Wall Area (X3), Roof Area (X4), Overall Height (X5), Orientation (X6), Glazing Area (X7), and Glazing Area Distribution (X8). License: CC-BY 4.0.

- **Bike sharing data** ($N = 17,379$; $d_X = 11, d_Y = 1$). The data was obtained from the UCI Machine Learning Repository (Fanaee-T, 2013). This was a regression task involves predicting the hourly count of rental bikes. The features included: season, year, month, hour, holiday, weekday, workingday, weather situation (weathersit), temperature (temp), apparent temperature (atemp), humidity (hum), and windspeed. License: CC-BY 4.0.

- **Air quality data** ($N = 9,358$). Sourced from the UCI Machine Learning Repository (Vito, 2008). License: CC-BY 4.0. We constructed two evaluation tasks:

  (i) *Univariate ($d_X = 5, d_Y = 1$):* Predict the PT08.S1(CO) sensor response using 5 other chemical sensor features ($N_{\text{train}} \approx 5,614$). This is our baseline real-world air quality experiment and is included in Section 4.

  (ii) *Multivariate ($d_X = 3, d_Y = 5$):* Predict 5 metal oxide sensor responses simultaneously using Temperature, Relative Humidity, and Absolute Humidity as inputs ($N_{\text{train}} \approx 5,400$). This setup tests the method's scalability to correlated multi-output regression. This experiment is detailed in the multidimensional ablations reported in Section 5.

- **Protein tertiary structure** ($N = 10,000$; $d_X = 9, d_Y = 1$). Data from the UCI ML Repository (UCI Machine Learning Repository, 2012). Regression to predict the protein backbone deviation (RMSD) from physicochemical descriptors. License: CC-BY 4.0.

- **NYC taxi trip duration** ($N = 100,000$; $d_X = 11, d_Y = 1$). Trip records released by the NYC Taxi & Limousine Commission (TLC) (NYC Taxi and Limousine Commission, 2016). The task is to predict *trip duration* (seconds) from pickup/dropoff coordinates and time features (hour, day, weekend/rush-hour indicators); we also include distance features (Haversine and Manhattan) and simple direction interactions. License: NYC Open Data Terms of Use.

- **Indoor localization data** ($N = 21,048$; $d_X = 520, d_Y = 2$). Sourced from the UCI Machine Learning Repository (Torres-Sospedra & Avariento, 2014). This is a spatial regression task to predict the exact position (Longitude, Latitude) of a user inside a multi-building campus based on the signal intensity (RSSI) of 520 WiFi access points. We utilized a random subset of 5,000 samples for our experiments ($N_{\text{train}} = 3,000$). This experiment is detailed in the multidimensional ablations reported in Section 5. License: CC-BY 4.0.

Table 5: Extended real-world comparison (coverage = mean; width = mean $\pm 1$ SD). Best width in **bold**, second best underlined, considering only methods with coverage $\geq 0.89$ per dataset.

| Dataset | Metric | AMDCP | DCP | CQR | SCP | ACP | EnbPI | LWCP |
|---|---|---|---|---|---|---|---|---|
| Energy | Coverage | 0.903 | 0.786 | 0.877 | 0.907 | 0.901 | 0.907 | 0.675 |
| | Width | **0.168 $\pm$ 0.081** | 0.381 $\pm$ 0.078 | 0.168 $\pm$ 0.015 | 4.824 $\pm$ 0.180 | 0.196 $\pm$ 0.050 | 4.808 $\pm$ 0.179 | 0.100 $\pm$ 0.008 |
| Bike | Coverage | 0.921 | 0.785 | 0.898 | 0.901 | 0.883 | 0.901 | 0.687 |
| | Width | 0.579 $\pm$ 0.449 | 0.481 $\pm$ 0.049 | **0.536 $\pm$ 0.012** | 4.696 $\pm$ 0.111 | 0.708 $\pm$ 0.215 | 4.300 $\pm$ 0.084 | 0.830 $\pm$ 0.037 |
| Air | Coverage | 0.904 | 0.789 | 0.881 | 0.900 | 0.895 | 0.900 | 0.778 |
| | Width | 0.941 $\pm$ 0.321 | 0.573 $\pm$ 0.051 | 0.712 $\pm$ 0.046 | 4.678 $\pm$ 0.322 | **0.839 $\pm$ 0.134** | 4.678 $\pm$ 0.321 | 0.668 $\pm$ 0.053 |
| Bio | Coverage | 0.931 | 0.946 | 0.898 | 0.899 | 0.881 | 0.904 | 0.906 |
| | Width | **2.125 $\pm$ 0.081** | 3.076 $\pm$ 0.049 | 2.208 $\pm$ 0.030 | 2.434 $\pm$ 0.048 | 2.322 $\pm$ 0.247 | 2.687 $\pm$ 0.052 | 2.500 $\pm$ 0.038 |
| Taxi | Coverage | 0.902 | 0.942 | 0.906 | 0.904 | 0.918 | 0.898 | 0.871 |
| | Width | **1.123 $\pm$ 0.040** | 3.785 $\pm$ 0.181 | 1.664 $\pm$ 0.092 | 1.487 $\pm$ 0.089 | 1.676 $\pm$ 0.371 | 1.681 $\pm$ 0.163 | 1.319 $\pm$ 0.074 |

### C.3.1 EXTENDED RESULTS

Table 5 shows that AMDCP attains target coverage while remaining competitive on width, tying CQR for the narrowest intervals on *Energy* and ranking second on *Bike*. On *Air*, CQR is narrowest among coverage-eligible methods, with ACP second. Methods with systematically wider regions (e.g., SCP/EnbPI) maintain coverage but at substantially higher width, whereas DCP often yields narrow intervals alongside undercoverage. Overall, AMDCP remains a strong real-world baseline under the $\geq 85\%$ coverage filter.

# D ABLATION STUDIES

## D.1 COMPONENT SENSITIVITY

This section provides the detailed results for our analysis of AMDCP's sensitivity to the hyperparameter $K$, the number of mixture components. We analyze three distinct scenarios: (1) performance on standard synthetic data, (2) a stress test for over-specification on simple unimodal data, and (3) a stress test for under-specification on highly complex, non-Gaussian data.

### D.1.1 PERFORMANCE ON STANDARD DATA

The results in Table 6 show that performance, in terms of both coverage and interval width, stabilizes quickly once $K$ is large enough to capture the underlying modes in the data (e.g., $K = 3$ or $K = 5$). Adding additional components beyond this point has a negligible effect on performance, indicating that for typical complex data, AMDCP is not sensitive to a moderate over-estimation of $K$.

Table 6: Sensitivity of coverage and interval width to the number of mixture components on standard synthetic multimodal dataset. Performance stabilizes quickly for $K \geq 3$.

| Components | Coverage | Target | Avg Width $\pm$ Std |
|:---:|:---:|:---:|:---:|
| 1 | 0.901 | 0.900 | $0.879 \pm 0.298$ |
| 2 | 0.900 | 0.900 | $0.820 \pm 0.291$ |
| 3 | 0.913 | 0.900 | $0.821 \pm 0.295$ |
| 5 | 0.905 | 0.900 | $0.807 \pm 0.288$ |
| 15 | 0.915 | 0.900 | $0.843 \pm 0.312$ |

### D.1.2 STRESS TESTS FOR SEVERE MODEL MISSPECIFICATION

To further probe the limits of AMDCP's robustness, we designed two stress tests focused on severe model misspecification where the chosen $K$ is far from optimal.

**Stress Test 1: Over-specification on Unimodal Data** Here, we test the scenario where the model is far more complex than the data. We generated a simple unimodal dataset (where the true $K = 1$) and intentionally over-specified the number of components for AMDCP up to $K = 100$. The results in Table 7 show that performance is highly stable. The MDN learns to assign near-zero mixture weights (typically $< 0.05$) to all but one component, effectively reducing itself to a simpler distribution. This self-regularizing behavior prevents significant degradation in prediction interval width, with only a 9.7% increase from $K = 1$ to $K = 100$, confirming AMDCP's robustness to over-specification.

Table 7: AMDCP performance on simple unimodal data (True $K = 1$) vs. the number of specified mixture components ($K$). The results demonstrate high robustness to over-specifying $K$.

| Components ($K$) | Coverage | Avg. Width |
|:---:|:---:|:---:|
| 1 (Oracle) | 0.901 | 1.619 |
| 2 | 0.898 | 1.646 |
| 5 | 0.899 | 1.660 |
| 10 | 0.898 | 1.673 |
| 25 | 0.918 | 1.834 |
| 50 | 0.900 | 1.760 |
| 100 | 0.904 | 1.917 |

**Stress Test 2: Under-specification on a Ring Distribution** Here, we test the opposite scenario, where the data is more complex than the model can theoretically represent with few components. We used a synthetic dataset where the conditional distribution follows a ring shape, a classic challenge for Gaussian Mixture Models which requires "tiling" the ring with many components. We intentionally under-specified AMDCP with only $K = 3$ and compared its performance against CQR and PCP.

The results in Table 8 show that even in this mismatched setting, AMDCP provides valid coverage (degrading by less than 5%) and, crucially, produces more efficient prediction regions than both baselines. This demonstrates that AMDCP degrades gracefully and maintains a competitive advantage even when its underlying model is misspecified.

Table 8: Performance on a ring-shaped distribution with a severely under-specified AMDCP ($K = 3$). Even when misspecified, AMDCP produces narrower intervals than existing methods.

| Method | Coverage | Avg. Width |
|--------|----------|-----------|
| CQR | 0.905 | 1.88 |
| PCP | 0.981 | 2.16 |
| AMDCP | 0.901 | 1.84 |

Experiments demonstrate that AMDCP is robust to number of mixture components. We recommend practitioners start with 3-5 components initially and our experiments indicate that good performance can be achieved over a range of K values, though optimal efficiency may require some tuning.

## D.2 ARCHITECTURE ABLATION

AMDCP's effectiveness is not heavily dependent on architectural choice, making it practical for deployment across diverse application domains without extensive architecture tuning.

Table 9: Comparison of architectures (1 trial, mean cov. and mean width $\pm$ std. dev.)

| Dataset | Metric | AMDCP Architecture | | | |
|---------|--------|------|-------------|------|--------|
| | | MLP | Transformer | LSTM | ResNet |
| Multimodal Data | Coverage | 0.902 | 0.896 | 0.888 | 0.914 |
| | Width | $2.033 \pm 0.653$ | $1.966 \pm 0.616$ | $2.004 \pm 0.732$ | $2.030 \pm 0.740$ |
| Mixture Data | Coverage | 0.892 | 0.893 | 0.887 | 0.895 |
| | Width | $2.147 \pm 0.793$ | $2.188 \pm 0.711$ | $2.238 \pm 0.897$ | $2.162 \pm 0.805$ |
| Skewed Heteroskedastic Data | Coverage | 0.913 | 0.904 | 0.904 | 0.923 |
| | Width | $1.519 \pm 0.586$ | $1.389 \pm 0.399$ | $1.430 \pm 0.448$ | $1.533 \pm 0.567$ |
| Regime-Switching Data | Coverage | 0.892 | 0.899 | 0.891 | 0.885 |
| | Width | $0.852 \pm 0.280$ | $0.919 \pm 0.309$ | $0.869 \pm 0.284$ | $0.853 \pm 0.293$ |
| Multidimensional Data | Coverage | 0.875 | 0.898 | 0.886 | 0.897 |
| | Width | $1.141 \pm 0.318$ | $1.175 \pm 0.328$ | $1.158 \pm 0.300$ | $1.225 \pm 0.371$ |

## D.3 DATASET SIZE

We vary the available training data from $10\%$ to $100\%$ (random splits per fraction), holding out a calibration set each time; results are averaged over multiple runs. Table 10 reports AMDCP coverage and average width (mean $\pm$ 1 SD) across five real datasets.

Coverage remains near the $90\%$ target once calibration sets are moderately sized, while widths shrink as data increases. Sensitivity is dataset-dependent: *Bio* and *Taxi* are relatively stable; *Energy* and *Bike* widen more at small fractions; *Air* is mild.

With moderate calibration sizes, AMDCP maintains $\approx 90\%$ coverage, and width generally decreases with more data; the degree of width contraction is dataset-dependent (most stable on *Taxi*, strongest gains on *Energy*/*Bike*).

## D.4 UNDER-FITTING AND OVER-FITTING

See Table 11 for experiment results.

Table 10: AMDCP stability under data subsampling (averaged over 5 runs).

| Dataset | Train fraction | $n_{\text{train}}$ (mean) | $n_{\text{cal}}$ (mean) | Coverage $(\mu \pm \sigma)$ | Avg. Width $(\mu \pm \sigma)$ |
|---|---|---|---|---|---|
| Air Quality | 0.10 | 49 | 17 | $0.859 \pm 0.091$ | $0.862 \pm 0.274$ |
| Air Quality | 0.25 | 123 | 42 | $0.849 \pm 0.052$ | $0.789 \pm 0.095$ |
| Air Quality | 0.50 | 247 | 83 | $0.861 \pm 0.042$ | $0.712 \pm 0.084$ |
| Air Quality | 0.75 | 371 | 124 | $0.853 \pm 0.063$ | $0.680 \pm 0.079$ |
| Air Quality | 1.00 | 495 | 166 | $0.904 \pm 0.003$ | $0.941 \pm 0.321$ |
| Bike Sharing | 0.10 | 1042 | 348 | $0.893 \pm 0.020$ | $2.085 \pm 0.014$ |
| Bike Sharing | 0.25 | 2606 | 869 | $0.895 \pm 0.005$ | $1.551 \pm 0.097$ |
| Bike Sharing | 0.50 | 5213 | 1738 | $0.889 \pm 0.004$ | $0.821 \pm 0.028$ |
| Bike Sharing | 0.75 | 7820 | 2607 | $0.903 \pm 0.008$ | $0.751 \pm 0.037$ |
| Bike Sharing | 1.00 | 10427 | 3476 | $0.921 \pm 0.012$ | $0.579 \pm 0.449$ |
| Bio | 0.10 | 600 | 200 | $0.881 \pm 0.019$ | $2.130 \pm 0.148$ |
| Bio | 0.25 | 1500 | 500 | $0.877 \pm 0.026$ | $1.998 \pm 0.157$ |
| Bio | 0.50 | 3000 | 1000 | $0.904 \pm 0.005$ | $1.935 \pm 0.052$ |
| Bio | 0.75 | 4500 | 1500 | $0.892 \pm 0.008$ | $1.962 \pm 0.059$ |
| Bio | 1.00 | 6000 | 2000 | $0.931 \pm 0.007$ | $2.125 \pm 0.081$ |
| Energy | 0.10 | 45 | 16 | $0.842 \pm 0.111$ | $0.943 \pm 0.343$ |
| Energy | 0.25 | 114 | 39 | $0.890 \pm 0.091$ | $0.879 \pm 0.262$ |
| Energy | 0.50 | 230 | 77 | $0.913 \pm 0.059$ | $0.835 \pm 0.177$ |
| Energy | 0.75 | 345 | 115 | $0.924 \pm 0.015$ | $0.556 \pm 0.025$ |
| Energy | 1.00 | 460 | 154 | $0.903 \pm 0.000$ | $0.168 \pm 0.081$ |
| Taxi | 0.10 | 5941 | 1981 | $0.901 \pm 0.004$ | $1.341 \pm 0.042$ |
| Taxi | 0.25 | 14854 | 4952 | $0.893 \pm 0.003$ | $1.308 \pm 0.049$ |
| Taxi | 0.50 | 29709 | 9903 | $0.905 \pm 0.002$ | $1.334 \pm 0.023$ |
| Taxi | 0.75 | 44563 | 14855 | $0.904 \pm 0.002$ | $1.274 \pm 0.007$ |
| Taxi | 1.00 | 59418 | 19806 | $0.902 \pm 0.003$ | $1.123 \pm 0.040$ |

Table 11: Impact of Training Misspecification (Overfitting and Underfitting) on AMDCP Performance. The results demonstrate that conformal calibration maintains the target coverage guarantee even when the underlying density model is poorly trained. However, misspecification leads to wider, less efficient prediction intervals compared to a well-trained model.

| Model Condition | Coverage | Avg. Width |
|---|---|---|
| *Scenario: Overfitting* | | |
| Well-Trained | $0.927 \pm 0.015$ | $2.692 \pm 0.051$ |
| Overfit | $0.927 \pm 0.016$ | $3.016 \pm 0.062$ |
| *Scenario: Underfitting* | | |
| Well-Trained | $0.901 \pm 0.017$ | $2.799 \pm 0.095$ |
| Underfit | $0.895 \pm 0.022$ | $3.074 \pm 0.166$ |

## D.5 EXTREME DATA DISTRIBUTIONS

Table 12: Performance under Distributional Misspecification (Target Coverage: 90%). Results are averaged over 5 runs $(\mu \pm \sigma)$.

(a) Asymmetric Skewed Data

| Method | Cov. | Width |
|---|---|---|
| AMDCP | $0.899 \pm 0.006$ | $\mathbf{0.987 \pm 0.032}$ |
| CQR | $0.895 \pm 0.017$ | $1.016 \pm 0.034$ |
| PCP | $0.995 \pm 0.002$ | $2.038 \pm 0.154$ |

(b) Symmetric Heavy-Tailed Data

| Method | Cov. | Width |
|---|---|---|
| AMDCP | $0.900 \pm 0.001$ | $\mathbf{4.268 \pm 0.001}$ |
| CQR | $0.898 \pm 0.012$ | $4.389 \pm 0.254$ |
| PCP | $0.994 \pm 0.003$ | $15.154 \pm 1.347$ |

As Table 12 shows, AMDCP successfully maintains strong performance with tighter intervals than CQR with comparable coverage. Crucially, PCP's prediction intervals suffer in these extreme distributions significantly more than AMDCP. This demonstrates the practical robustness of AMDCP especially in comparison to generative models, in data with extreme distributions.

### D.6 SCALABILITY ON MULTIDIMENSIONAL DATA

To evaluate the scalability of AMDCP to multivariate regression tasks, we conducted experiments on both synthetic and real-world high-dimensional datasets.

**Synthetic Mixture Data.** We generated a dataset with fixed input dimension $d_X = 5$ and variable output dimension $d_Y \in \{2, 5, 10, 20\}$. The data generating process simulates a complex joint distribution with three distinct functional modes: (1) a linear trend $Y = XW_1 + \epsilon$; (2) a saturation trend $Y = 4\tanh(XW_2) + \epsilon$; and (3) a sinusoidal trend $Y = 3\sin(XW_3) + \epsilon$. We also introduced autoregressive correlations between output dimensions, where $Y_d \leftarrow Y_d + 0.25Y_{d-1}$. This results in a target distribution that is both multimodal and highly correlated across dimensions. We do the same for **Synthetic Multimodal Data**.

**Real-World Multivariate Data.** We utilized two datasets:

1. **UCI Air Quality** ($d_Y = 5$)**:** Predicting responses of 5 metal oxide sensors simultaneously based on atmospheric inputs.

2. **UCI Indoor Localization** ($d_Y = 2$)**:** Predicting spatial coordinates (Latitude, Longitude) from WiFi signal strengths.

Table 13 (Synthetic) and Table 14 (Real-World) summarize the results. AMDCP demonstrates robust scaling. On synthetic data, it consistently outperforms all baselines. On real-world data, it achieves the best efficiency on Air Quality and ties for best on Indoor Localization with CQR. Crucially, AMDCP avoids the volume explosion observed in generative baselines (PCP, DCP), which tend to over-cover significantly in high dimensions.

Table 13: Scalability analysis on synthetic high-dimensional data ($N = 1000$). Target coverage is 0.90. Best width in **bold**, second best underlined.

| Scenario | Dim ($d_Y$) | Metric | AMDCP | CQR | SPCI | CONTRA | PCP | DCP |
|---|---|---|---|---|---|---|---|---|
| Panel A: Mixture Data | 2 | Coverage | 0.91 | 0.91 | 0.82 | 0.90 | 0.91 | 1.00 |
| | | Width | **4.93** | 8.49 | 12.20 | 11.81 | 13.27 | 13.26 |
| | 5 | Coverage | 0.91 | 0.90 | 0.63 | 0.90 | 0.90 | 1.00 |
| | | Width | **7.68** | 8.76 | 14.62 | 14.04 | 18.18 | 18.17 |
| | 10 | Coverage | 0.91 | 0.90 | 0.46 | 0.90 | 0.90 | 1.00 |
| | | Width | **10.61** | 11.96 | 17.71 | 16.76 | 23.89 | 23.95 |
| Panel B: Multimodal Data | 2 | Coverage | 0.89 | 0.89 | 0.69 | 0.90 | 0.89 | 0.99 |
| | | Width | **2.48** | 2.77 | 5.03 | 5.97 | 4.81 | 4.80 |
| | 5 | Coverage | 0.89 | 0.90 | 0.54 | 0.91 | 0.90 | 1.00 |
| | | Width | **4.84** | 6.30 | 6.24 | 7.37 | 7.82 | 7.81 |
| | 10 | Coverage | 0.90 | 0.90 | 0.41 | 0.89 | 0.90 | 1.00 |
| | | Width | **7.85** | 12.91 | 8.40 | 9.81 | 13.33 | 13.28 |

Table 14: Real-world multivariate comparison (coverage = mean; width = mean $\pm 1$ SD). Best width in **bold**, second best underlined, considering methods with coverage $\approx 0.90$.

| Dataset | Metric | AMDCP | **CONTRA** | CQR | DCP | PCP | SPCI |
|---|---|---|---|---|---|---|---|
| Air Quality ($d_Y = 5$) | Coverage | 0.911 | 0.904 | 0.904 | 0.998 | 0.998 | 0.910 |
| | Width | **1092.01 $\pm$ 0.58** | 1178.86 $\pm$ 5.15 | 1411.51 $\pm$ 0.32 | 2163.41 $\pm$ 13.77 | 2151.23 $\pm$ 6.90 | 1321.23 $\pm$ 0.01 |
| Indoor Loc ($d_Y = 2$) | Coverage | 0.898 | 0.899 | 0.899 | 0.973 | 0.970 | 0.891 |
| | Width | 33.82 $\pm$ 0.01 | 71.09 $\pm$ 0.01 | **33.76 $\pm$ 0.01** | 53.18 $\pm$ 13.53 | 51.14 $\pm$ 6.76 | 71.19 $\pm$ 0.01 |

# E    CONDITIONAL COVERAGE ANALYSIS

While perfect conditional coverage is theoretically impossible in finite samples without distributional assumptions, we analyze how coverage varies across different regions of the feature space as a practical measure of conditional reliability.

## E.1    CONDITIONAL COVERAGE EXPERIMENTS

**Binned conditional coverage analysis**    Test data was partitioned into 10 quantile bins based on the primary feature (for 1D data) or Euclidean distance/principal feature (for High-D data). Table 15 reports the average and worst-bin coverage. AMDCP consistently improves worst-bin coverage compared to SCP, particularly in multimodal or complex structures. This demonstrates AMDCP's superior ability to adapt to local data manifolds, providing more uniform uncertainty quantification in regions where traditional methods often under-cover.

Table 15: Conditional coverage analysis on synthetic datasets. 'Improvement over SCP (Worst-Bin, pp)' shows the absolute percentage point increase in worst-bin coverage compared to the SCP baseline. Best worst-bin coverage for each dataset is in **bold**.

| Dataset | Method | Target Cov. | Avg. Cov. (Empirical) | Worst-Bin Cov. | Improvement over SCP (Worst-Bin, pp) |
|---|---|---|---|---|---|
| Multimodal | AMDCP | 90.0% | 92.0% | **87.5%** | +24.5 |
|  | SCP | 90.0% | 89.6% | 63.0% | - |
| Mixture | AMDCP | 90.0% | 87.1% | **82.0%** | +17.5 |
|  | SCP | 90.0% | 90.0% | 64.5% | - |
| Skewed Heterosk. | AMDCP | 90.0% | 90.6% | **82.5%** | +6.0 |
|  | SCP | 90.0% | 90.3% | 76.5% | - |
| Regime-Switching | AMDCP | 90.0% | 89.2% | **81.5%** | +18.0 |
|  | SCP | 90.0% | 91.6% | 63.5% | - |
| Multidimensional | AMDCP | 90.0% | 89.8% | **87.5%** | +35.0 |
|  | SCP | 90.0% | 91.5% | 52.5% | - |

**Coverage on random euclidean balls (COV-EUCS)**    To further assess local coverage properties, we evaluated coverage within random Euclidean balls in the feature space (COV-EUCS). For each dataset, we sampled 100 test points to serve as centers for these balls. The radius for each ball was determined heuristically based on a chosen percentile (typically 10th or 20th, see Table 16) of the pairwise Euclidean distances among a random subset of 1000 test points. The results indicate that AMDCP's adaptive nature helps maintain coverage more consistently in localized regions of the feature space where simpler models might falter. The average number of points per ball indicates that these local neighborhoods are non-trivial.

## E.2    APPROXIMATE CONDITIONAL COVERAGE OF BASELINE AMDCP

A rigorous, finite-sample, individual-conditional coverage guarantee is generally impossible to achieve distribution-free without knowing the true data-generating process (Vovk, 2012). However, approximate conditional coverage can be achieved if the non-conformity score $s(X, Y)$ has a distribution that is nearly invariant to the conditioning variable $X$.

Following the framework established by Chernozhukov et al. (2021) and Lei et al. (2018), such invariance is maximized when the score approximates the true conditional density, i.e., $s(x, y) \approx -\log p(y|x)$. AMDCP leverages this principle by training a MDN to estimate $p(y|x)$ and defining the non-conformity score via the weighted Mahalanobis distance to the dominant component. To the extent that the MDN successfully approximates the true data-generating process, the distribution of these scores becomes orthogonal to $X$, ensuring that the global calibration threshold $\hat{q}_{1-\alpha}$ provides valid coverage locally across the feature space. We refer the reader to Chernozhukov et al. (2021) for the complete argument. This argument establishes that AMDCP is designed on sound principles for achieving conditional coverage. The empirical results in Appendix E support this, showing that AMDCP consistently achieves better worst-case local coverage than methods with simpler scores.

Table 16: Approximate x-conditional coverage using random euclidean balls (COV-EUCS). Best worst-ball coverage for each dataset is in **bold**.

| Dataset | Method | Avg. Ball Cov. | Worst-Ball Cov. | Median Ball Cov. | Avg. Points per Ball | Radius Perc. Used (%) |
|---|---|---|---|---|---|---|
| Multimodal (Synth) | AMDCP | 88.8% | **77.7%** | 88.8% | 99.0 | 10 |
| | SCP | 89.5% | 62.9% | 95.7% | 103.6 | 10 |
| Mixture (Synth) | AMDCP | 91.2% | **83.7%** | 91.8% | 100.5 | 10 |
| | SCP | 88.2% | 47.5% | 95.3% | 99.3 | 10 |
| Skewed Heterosk. (Synth) | AMDCP | 90.3% | 78.5% | 90.7% | 100.9 | 10 |
| | SCP | 91.8% | **79.7%** | 91.2% | 99.9 | 10 |
| Regime-Switching (Synth) | AMDCP | 91.1% | **83.5%** | 90.9% | 102.4 | 10 |
| | SCP | 91.5% | 53.5% | 98.1% | 102.1 | 10 |
| Multidimensional (Synth) | AMDCP | 90.0% | **82.6%** | 89.4% | 104.9 | 10 |
| | SCP | 88.9% | 54.9% | 92.7% | 102.3 | 10 |
| Energy Efficiency | AMDCP | 85.2% | **70.8%** | 85.7% | 32.8 | 20 |
| | SCP | 91.3% | 68.8% | 95.5% | 32.2 | 20 |
| Bike Sharing | AMDCP | 91.1% | **69.2%** | 91.7% | 744.3 | 20 |
| | SCP | 90.8% | 65.7% | 91.4% | 745.6 | 20 |
| Air Quality | AMDCP | 85.2% | 68.4% | 85.7% | 21.2 | 10 |
| | SCP | 90.3% | **70.0%** | 91.6% | 22.2 | 10 |

### E.3 GROUP-CALIBRATED AMDCP FOR CONDITIONAL COVERAGE

Standard conformal prediction guarantees marginal coverage, $P(Y_t est \in C(X_t est)) \geq 1 - \alpha$. However, in many high-stakes applications, it is critical to ensure coverage conditional on specific subgroups. To address this, we propose Group-Calibrated AMDCP (G-AMDCP).

**Methodology.** G-AMDCP operates in settings where data points $(X_i, Y_i)$ are accompanied by a discrete group label $G_i \in \{1, ..., M\}$. The core idea is to perform the calibration step of the conformal prediction algorithm independently for each group. Given a test point $X_{test}$ with group label $G_{test}$, the non-conformity scores of the calibration points are filtered to only include those belonging to the same group, $D_{cal}^{G_{test}} = \{(X_i, Y_i) \in D_{cal} | G_i = G_{test}\}$. A separate quantile $\hat{q}_{1-\alpha}^{G_{test}}$ is then computed from this subgroup's scores. This yields a prediction region that provides a sharp, finite-sample coverage guarantee conditional on group membership: $P(Y_{test} \in C(X_{test}) | G_{test} = g) \geq 1 - \alpha$ for each group $g$.

**Experimental Validation.** We tested G-AMDCP on a synthetic dataset with a complex minority subgroup. As shown in Table 17, standard methods that achieve marginal coverage exhibit severe under-coverage for the minority group. In contrast, G-AMDCP successfully achieves the target 90% coverage for both the majority and minority groups without sacrificing efficiency.

Table 17: Performance of G-AMDCP on the Asymmetric Group-Conditional dataset (Target Coverage 90%). G-AMDCP provides valid conditional coverage for both groups.

| Method | Avg. Width | Marginal Cov. | Group 0 (n=824) Cov. | Group 1 (n=176) Cov. |
|---|---|---|---|---|
| *Standard (Marginal) Methods* | | | | |
| Standard SCP | 3.316 | 0.904 | 0.995 | 0.531 |
| Standard AMDCP | 1.842 | 0.906 | 0.919 | 0.852 |
| **G-AMDCP** | **1.899** | **0.911** | **0.907** | **0.929** |

### E.4 THEORETICAL ANALYSIS OF CONDITIONAL COVERAGE

When discrete group information $G$ is available, we can move from an approximate argument to a rigorous, provable guarantee. The G-AMDCP procedure, defined in Appendix E.3, achieves this by performing calibration independently within each group. For brevity, we present brief proofs below. The procedure performs split-conformal calibration independently within each group $g$, which is a structured application of localized conformal prediction principles (Guan, 2020; Tibshirani et al.,

2019) which cotnained detailed proofs of these properties. When discrete group information $G$ is available, G-AMDCP moves from approximate guarantees to rigorous finite-sample validity.

**Theorem E.1** (Group-Conditional Coverage Guarantee). *Let $\{(X_i, G_i, Y_i)\}$ be an exchangeable sequence. For any group $g \in \{1, \ldots, M\}$ such that the calibration subset $D_{cal,g}$ is non-empty, the prediction region $C_\alpha$ constructed by G-AMDCP satisfies:*

$$Pr\left(Y_{test} \in C_\alpha(X_{test}, G_{test}) \mid G_{test} = g\right) \geq 1 - \alpha$$

*Proof.* Conditioning on the event $G_i = g$ for all $i$ in the calibration subset and the test point effectively defines a subsequence of observations. Since the original sequence is exchangeable, any subsequence defined by a deterministic rule (here, group membership) retains exchangeability conditional on the group labels. The result then follows directly from the standard marginal coverage guarantee of split-conformal prediction applied to this subsequence (Vovk, 2012, Lemma 1). □

**Theorem E.2** (Marginal Coverage Guarantee of G-AMDCP). *Under the same assumptions, G-AMDCP also satisfies the marginal coverage property:*

$$Pr\left(Y_{test} \in C_\alpha(X_{test}, G_{test})\right) \geq 1 - \alpha$$

*Proof.* This follows immediately from the Law of Total Probability. By Theorem E.1, the coverage probability conditional on any specific group $g$ is at least $1 - \alpha$. Averaging these conditional probabilities over the distribution of groups $\Pr(G_{test} = g)$ yields a marginal probability of at least $1 - \alpha$. □

# F  GUIDE FOR PRACTITIONERS

This guide provides practical advice for deploying AMDCP, addressing common questions about its application, tuning, and trade-offs. Our recommendations are grounded in the extensive empirical evaluations and stress tests detailed throughout the paper.

## F.1  WHEN SHOULD I USE AMDCP?

AMDCP's primary advantage lies in its ability to flexibly model complex conditional distributions. The decision to use it over simpler methods should be guided by the nature of your data.

- **Use AMDCP when you suspect or require:**

    - **Multimodality:** The outcome $y$ could plausibly have multiple distinct values for a single input $x$. For example, predicting the future position of a pedestrian at an intersection (they could go left, right, or straight).

    - **Complex Heteroskedasticity:** The uncertainty of the prediction changes in non-trivial ways with the input features, not just simple scaling.

    - **Non-Contiguous Prediction Regions:** The plausible regions of high probability are naturally disjoint, and a single connected interval would be misleadingly large.

- **When to Prefer Simpler Methods:** For datasets that are strongly unimodal and homoskedastic, the extra modeling capacity of AMDCP may not be necessary. While AMDCP still provides leading performance or exceeds current standards, a simpler model may suffice.

**Diagnostic Steps.**  To determine if AMDCP is the right tool, we recommend the following:

1. **Train a Simple Baseline:** Fit a standard regression model (e.g., OLS, Gradient Boosting).

2. **Analyze the Residuals:** Plot the residuals ($y_{\text{true}} - y_{\text{pred}}$) against the predicted values or key input features. If you observe distinct clusters, multiple horizontal "bands" of errors, or complex funnel shapes, these are strong indicators of multimodality or heteroskedasticity where AMDCP will excel.

    For example, we see below the residuals of the energy efficiency dataset for both OLS and a gradient boosted regressor:

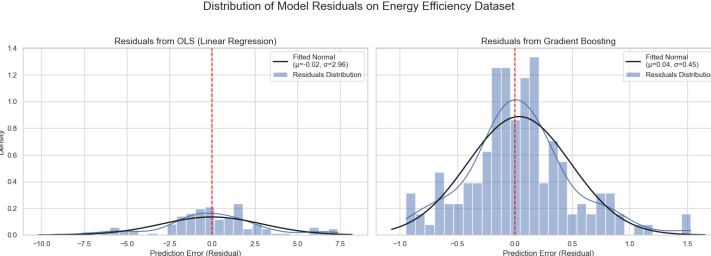

3. **Statistical Tests:** We also recommend utilizing statistical tests for multimodality and/or heteroskedasticity, such as Hartigan's Dip Test, the Mclust BIC Comparison, and the Breusch-Pagan Test. We present example results below for a few of the real-world datasets tested in Section 5.

Table 18: Statistical diagnostics for Multimodality and Heteroskedasticity on real-world datasets. 'Dip Test' $p < 0.05$ suggests multimodality. 'Optimal $K_{BIC}$' $> 1$ indicates a mixture model is preferred. 'Breusch-Pagan' $p < 0.05$ indicates heteroskedasticity. All datasets show strong heteroskedasticity, and most require mixture modeling.

| Dataset | Multimodality Evidence | | Heteroskedasticity | Distributional Conclusion |
|---|---|---|---|---|
| | Dip Test ($p$-val) | Optimal $K$ (BIC) | Breusch-Pagan ($p$-val) | |
| Energy Efficiency | **0.000** | 2 | **0.000** | Multimodal & Heteroskedastic |
| Bike Sharing | **0.000** | 3 | **0.000** | Multimodal & Heteroskedastic |
| Protein (Bio) | **0.000** | 3 | **0.000** | Multimodal & Heteroskedastic |
| Air Quality | 0.989 | 2 | **0.000** | Heteroskedastic (Mixture Pref.) |
| NYC Taxi | 0.926 | 3 | **0.000** | Heteroskedastic (Heavy-Tailed) |

4. **Consider the Domain:** In fields where the underlying process is known to be multimodal (e.g., robotics, particle physics, certain economic forecasts), AMDCP is a strong candidate from the outset.

## F.2 How Do I Choose the Number of Components (K)?

A key hyperparameter is K, the number of mixture components. Our extensive stress tests (Appendix D.1) show that **AMDCP is highly robust to this choice**.

- **Recommended Starting Point:** A value of $K \in [3, 5]$ is a robust starting point for most applications and often requires no further tuning.

- **Over-specification is Safe:** Our experiments demonstrate that even when K is severely over-specified (e.g., using $K = 100$ on unimodal data), performance degrades minimally. The model learns to ignore extraneous components by assigning them near-zero mixture weights ($\pi_k \approx 0$), resulting in only a small increase in interval width while maintaining target coverage.

- **Under-specification Degrades Gracefully:** If K is too small to capture the true data complexity, AMDCP still produces valid coverage, and its intervals remain highly competitive, often tighter than leading alternatives like PCP and CQR even when misspecified.

This robustness mitigates concerns about extensive hyperparameter tuning, making AMDCP practical for real-world deployment where the true data complexity is unknown.

## F.3 What Kind of Coverage Guarantee Do I Get?

Understanding the type of coverage is critical for responsible deployment, especially in fairness-sensitive applications.

- **Marginal Coverage (Default):** Standard AMDCP provides a rigorous, finite-sample guarantee of **marginal coverage**. That is, averaged across all test points, the procedure will cover the true outcome at least $1 - \alpha$ of the time. This guarantee does *not* extend to specific subgroups.

- **Group-Conditional Coverage (Recommended for Fairness):** If your application requires formal coverage guarantees for specific, known subgroups (e.g., ensuring reliability across demographic groups, patient cohorts, or device types), you should use our proposed extension, **G-AMDCP** (see Appendix E.3). G-AMDCP performs calibration independently for each subgroup, providing a sharp, finite-sample guarantee of $\mathbb{P}\{Y \in \mathcal{C}(X) \mid G = g\} \geq 1 - \alpha$ for every group $g$.

## F.4 Computational Trade-Offs and Model Flexibility

- **Training Time:** The training cost of AMDCP is significantly lower than other advanced density-based methods. For instance, AMDCP trains approximately **2-3x faster** than normalizing flow models like CONTRA.

- **Inference Time:** This is AMDCP's key advantage. Because its prediction region is an analytic union of ellipsoids, membership testing is constant-time. It is consistently an **order of magnitude faster** (e.g., ∼17x faster on the Bio dataset) than sampling-based methods that produce non-contiguous regions, like PCP. This makes AMDCP suitable for real-time or large-scale batch processing.

- **Model Flexibility:** The MDN should be viewed as a lightweight output "head," not a restrictive architectural choice. This head can be attached to any feature-extracting backbone (MLP, Transformer, ResNet, etc.). Practitioners can leverage powerful **pre-trained models** by freezing the backbone and training only the MDN head, saving significant computational cost while gaining the benefits of our adaptive "glass-box" score.

### F.5 KNOWN LIMITATIONS AND FAILURE CASES

Based on our robustness analysis, practitioners should be aware of the following:

- **Extremely Low Data Regimes:** Like any method relying on a neural network, AMDCP's performance can degrade when the training set is extremely small, as the MDN may fail to learn the conditional density accurately. In such low-data scenarios, simpler, non-parametric methods may be more stable.

- **Distribution Shift:** While standard split-conformal methods assume exchangeability, AMDCP can be adapted for time-series data as discussed in Appendix A.1. However, under severe and abrupt distribution shifts, the calibration can become stale. We recommend periodic re-calibration as a standard best practice in production environments. This is outside the scope of this paper.

- **Optimization Complexity**: Training MDNs involves a non-convex loss landscape that can be more sensitive to initialization and hyperparameter tuning (e.g., number of components $K$) compared to simpler methods. While our ablations show robustness to $K$, convergence is not as guaranteed as in convex formulations.

- **Approximation Gap**: Our analytic score relies on the "max-component" approximation. While Corollary 4.1.1 proves asymptotic convergence under separation, in regimes with heavily overlapping modes, the "min" operator produces sharp geometric cusps at the intersection of ellipsoids, whereas the true density creates smooth valleys. This results in a slight geometric mismatch in the overlap regions compared to exact likelihood methods, representing the trade-off for achieving $O(1)$ inference speed.

- **Geometric Constraints**: AMDCP assumes the high-density regions can be efficiently tiled by ellipsoids. Although our ablations do indeed show robustness to distributions with highly irregular, non-convex shapes that are not well-approximated by Gaussian mixtures, flow-based methods may provide tighter fits, albeit at much higher training and inference costs.

