# OpenReview forum: "AMDCP: Adaptive Mixture Density for Conformal Prediction"
_ICLR.cc/2026/Conference — Submitted to ICLR 2026_

### Official Review · Reviewer_fHY2 · 2025-10-17

**Soundness:** 1
**Presentation:** 2
**Contribution:** 3
**Rating:** 4
**Confidence:** 4

**Summary:**

The authors propose AMDCP, a conformal prediction method based on mixture density networks. The method is invariant with respect to the underlying model architecture and easily allows for generating non-continuous prediction sets because score evaluation is only performed with the maximum density Gaussian component. Empirical results show that AMDCP works effictively and generates interpretable prediction sets.

**Strengths:**

Overall, I think the work is somewhat interesting.

1. The method is simple and architecture-invariant.

2. The method allows for easily generating discontinuous prediction sets.

3. The experimental evaluation is strong.

**Weaknesses:**

While the method is interesting, I also do not think it is particularly stellar. The following two aspects, especially the first one, are keeping me from giving a "borderline accept" score. If the authors successfully resolve 1. and improve on 2., I am willing to increase my score to "borderline accept".

1. The paper has a fundamental flaw in its theoretical contribution, namely Theorem 4.2: First of all, the proof is not very rigorous and it seems there are some implicit assumptions that may actually not hold for AMDCP. I am specifically worried about step 3, because AMDCP does not construct the non-conformity score as a function of the full density estimate, but does this max Gaussian approximation. The second problem is that the theorem statement is implied by the assumptions alone and is entirely disconnected from the proposed AMDCP. Any method that fulfills assumption (i) (in addition to some more assumptions that I believe are missing), should achieve the property. Assumption (ii), on the other hand, is trivially the case for any consistent estimate of the quantile (like the one used in conformal prediction). Perhaps there is a great miss-understanding from my side: In that case, I would be happy if the authors could clarify.

2. The style of writing is clear, but overall too grandiose. I suggest to adopt a more scientific style of writing. Just one example: *"Crucially, our framework retains exceptional modularity:"*. Words like *"exceptional"* can be safely dropped. On the same note, I believe that the appendix should be greatly reduced. I believe that the paper should only cover points that are either contributions or improve clarity, including the appendix. I therefore suggest referencing the original works for lengthy proofs. This will improve readability and clarify the contributions of the manuscript.

3. The method requires training a mixture density network head on top of an existing model, which may be prohibitive.

**Questions:**

* What is a "glass-box framework"? I have never heard this term before.

* Theorem 4.1: Would it be possible to stress more clearly that this result is well-known and comes from another work? For instance, I recommend *Theorem 4.1 (Marginal coverage guarantee; INSERT CITATION)*. This way, it is more clear that this theorem is not a contribution of this work. Personally, I am generally not a great fan of stating existing theorems like that because it makes it hard to see what the contribution is. I am aware that doing so is common these days, but I believe we should be clear about what our contributions are.

* 6 Discussion: The authors use the word "SOTA" a number of times. I believe this term is unscientific and should be avoided. I suggest a more balanced perspective that highlights upsides and downsides of the proposed method. It would be great if the authors could move Appendix F.1 (which has such a very useful analysis) to the main text and remove these uninformative SOTA claims.

* There are many well-known results and trivialities in the appendix. For instance, the proof for B.1 is well-known and I do not think it needs to be repeated in that length. Also, B.2.2 is just an existing result without any addition and I do not see why it has anything to do with AMDCP. Of course, it is all just in the appendix and the original works are properly cited, so it is not a great issue. Nevertheless, I have doubts whether we need to extend a paper to 30 pages just by listing well-known and uninformative results.

---

> ### Author Response · Authors · 2025-11-26
> **Discussion 1/2**
>
> Dear Reviewer fHY2,
>
> ### W1: Theoretical Constructions
>
> We thank the reviewer for the sharp eye on the theoretical construction. We have reframed the theoretical analysis in Section 4 and the proofs therein to clarify this and provide a more rigorous understanding of AMDCP.
>
> * We replaced the generic distribution convergence assumption with Parameter Consistency (A1) and used the Continuous Mapping Theorem and Dominated Convergence Theorem to prove that parameter convergence implies volume convergence for our specific union-of-ellipsoids geometry (Theorem 4.1). We have improved the theory sections to make this link explicit.
> * We proved a new Corollary 4.1.1 (and Appendix B.4) which explicitly bounds the difference between the true likelihood and our 'min' score. We show this difference is bounded by 2logK globally and vanishes asymptotically under a reasonable seperation condition (A4).
>
> If you have any additional feedback on how to better frame our theoretical contributions, please let us know and we would be happy to work with you to further improve our paper!
>
> ### W2, Q3, Q4: Writing Style
>
> We understand that the use of "SOTA" and other phrases may have come across as grandiose, and that was not our intention; we apologize for this. We have reduced this language in our revised manuscript and removed "SOTA" entirely. To streamline the paper, we have removed the detailed derivations and instead cited the original works where these results are established (See response to Q2). We also included a few lemmas to help clarify the proofs of our revised Section 4. We kept page breaks between appendix sections to improve readability which increased length. We are also intending to replace our full detailed proofs of conditional coverage in Appendix E with brief proof sketches, for brevity as they are not central to the manuscript.
>
> ### W3: Training an MDN Imposes Extra Costs
> We see the use of an MDN not as a restrictive assumption, but as a powerful and flexible modeling choice that enables our method to address a key limitation of many existing CP approaches: generating non-contiguous prediction intervals for potentially multimodal data, and doing so in an efficient (both interval length and inference-time) manner.
>
> You raise a point about the model's requirements, but we argue they are far less restrictive than they appear. Crucially, due to the universal approximation properties of such mixtures [1], this is an exceptionally flexible assumption, allowing MDNs to capture arbitrary conditional distributions. AMDCP's fast inference time and strong empirical results would enable high performance in real-time, allowing CP to be used in a variety of real-world applications (See Literature Review). This real-time CP with top-tier performance would be critical in a wide variety of fields.
>
> In terms of further training a model, as you point out, in the scenario where a user has a pre-trained feature extractor (e.g., a ResNet or Transformer backbone) and wishes to leverage it. In this case, AMDCP is indeed highly applicable. The user would freeze the pre-trained backbone and simply train a new, lightweight MDN "head" on top of the extracted features. This approach allows one to gain the benefits of our "glass-box" score for multimodal uncertainty without the cost of training a large model from scratch. This retains the spirit of leveraging pre-trained models while unlocking a much higher level of performance for complex distributions.
>
> **And in comparison to existing high-performance methods, which use heavier-requirement intensive models like CONTRA, our lightweight methodology trains in a third of the time, while outperforming CONTRA both in coverage, interval width, and inference time. And while AMDCP requires an MDN output layer, it retains significant flexibility in all previous layers; as our ablations confirm, it can be paired with any neural backbone (e.g., MLP, Transformer, ResNet as our ablations show), allowing it to adapt to diverse data structures, maintain rapid inference time performance, and train quicker than existing methods while retaining a high degree of flexibility.** This is also important as this means that the order of magnitude faster run times will hold across model architectures, enabling AMDCP to be used in a wide variety of specialized domains with specialized backbone architectures.
>
> Therefore, we see the MDN not as a restrictive requirement, but as a powerful tool that enables a more sophisticated class of non-conformity scores. This is precisely the capability that methods with "black-box" approaches, like DCP, lack, limiting them to simpler, contiguous prediction regions on data with complex heteroskedasticity or other distributional effects. By designing a score that leverages this unlocked structure, AMDCP provides a novel and robust contribution to the literature, offering a practical solution for complex data. Our empirical analysis further validates this conclusion.

---

> ### Author Response · Authors · 2025-11-26
> **Discussion 2/2**
>
> ### Q1: Glass-Box Explanation
>
> We simply use the glass-box phrase as a method to explain the innovation of AMDCP. Our primary innovation is a paradigm shift from treating generative models as "black-box" oracles to a "glass-box" approach that leverages their internal, learned structure. To our knowledge, AMDCP is the first method to design a non-conformity score that is an analytic function of a model's decomposed components—the means, covariances, *and* data-dependent mixing weights of an MDN which cannot be done by other methods, nor is done by current methods which use MDNs.
>
> Existing probabilistic methods like DCP and PCP treat the model as a black-box density oracle, using the total log-likelihood as their score. This forces the prediction region to be a single, connected super-level set, which is highly inefficient for multimodal data (as seen in Figure 1). Inside this "black box," the crucial structural information from the individual mixture components is effectively "averaged out" and lost in the final, aggregated density value. To instead generate non-contiguous regions, methods such as PCP thus use costly post-processing sampling methods, which is inefficient and costly.
>
> In sharp contrast, our score "opens the box." It leverages the MDN's internal structure through two key mechanisms:
>
> 1. **Selection over Aggregation:** The $min_k$ operator allows the creation of analytic, non-contiguous prediction regions (a union of ellipsoids).
> 1. **Learned Penalization:** The $\pi_k(x)$ term actively prunes improbable modes based on the network's own learned gating. In existing methods which use an MDN, their "black-box" nature means that while the mixture weights are mathematically present the final, aggregated density output obscures their individual roles. The crucial information about which mode is most probable for a given $x$ is effectively "averaged out" and lost. Our score, in contrast, directly leverages the mixing weights term to actively discourage the inclusion of components that the MDN deems implausible for a given input x, and shrinks the prediction regions around the most probable ones—a selective, adaptive mechanism that is impossible when the model's internal structure is ignored.For example, although PCP is an existing generative method to form non-contiguous regions, its "black-box" treatment of generative models forces it to form non-contiguous regions in a costly manner by generating samples and defining balls around test points.
>
>
> **The benefits of a glass-box approach:** This glass-box design directly leads to narrower, more adaptive regions and dramatically faster inference compared to other methods that produce non-contiguous sets.
>
> Our work demonstrates that by designing a score tailored to the model's analytic structure, one can achieve superior adaptivity without sacrificing computational efficiency. AMDCP's "glass-box" approach is highly competitive and often times better than other methods, achieving narrower intervals at comparable coverage with a significant advantage in inference-time computational efficiency (e.g., up to 17x faster than PCP, and over a 20\% speed up from CONTRA) and nearly 2x faster than CONTRA at train time. we see this as evidence that the specific "glass-box" way AMDCP leverages the MDN's structure via our novel non-conformity score provides a significant, practical advantage. We believe this novel mechanism, which unlocks prediction regions that are simultaneously adaptive, non-contiguous, and computationally efficient, represents a significant and practical contribution.
>
>
> ### Q2: Clarification of Theorem Contributions
>
> Although we had tried to do this originally, we thank you for your feedback that these connections were not stressed enough. In our revised manuscript, we further emphasize these other papers.

---

> > ### Comment · Reviewer_fHY2 · 2025-11-26
> >
> > I thank the authors for the response and revisions. I checked the revised theoretical results and they seem correct to me now (even though I believe the same result can be shown for other methods, too). Unfortunately, however, the authors have failed to follow my suggestion of properly outlining the limitations and downsides of their approach, in the main text. I believe that good science openly discusses these aspects. I will therefore keep my score.

---

> ### Author Response · Authors · 2025-11-26
>
> Dear Reviewer fHY2,
>
> We sincerely thank you for your continued engagement and for verifying the correctness of our revised theoretical results.
>
> ## On Theory
> We appreciate your acknowledgment that the results are correct. To clarify the intent of Theorem 4.1 and Corollary 4.1.1: Our goal was not merely to re-prove standard CP validity. Rather, we prove that our specific architectural choice—using the "Glass-Box" (max-component) approximation for fast training and rapid inference—converges asymptotically to the optimal set, like the traditional "Black-Box" (full log-likelihood) oracle.
>
> These theoretical results are crucial because they justify our method's core value proposition: they prove that we can achieve the computational efficiency, geometric tractability, and robustness of an analytic score without sacrificing the statistical optimality of a full density estimator. And in fact, our experiments go further by demonstrating that in finite samples (ie real-world data) this glass-box approach often achieves better performance, while also being extremely robust to a variety of ablations common to real-world datasets.
>
> ## On Weaknesses
> We take your comment regarding the discussion of limitations very seriously-you are absolutely right that good science requires openly discussing downsides. We had previously included a small discussion of tradeoffs in the main paper and kept the rest in the appendix but we apologize for not making this sufficiently prominent in the previous revision. Based on your feedback, we have revised the Discussion (Section 6) in the main paper to include a dedicated paragraph explicitly detailing the trade-offs of AMDCP. We believe this new paragraph provides the balanced perspective you requested, clearly distinguishing where AMDCP excels and where it faces friction.
>
> We have also extended our dedicated section "F.5 Known Limitations and Failure Cases" in the Appendix for an even deeper dive into failure modes.
>
> We hope these revisions demonstrate our commitment to a balanced and rigorous presentation of AMDCP.

---

> > ### Comment · Reviewer_fHY2 · 2025-11-27
> >
> > I have checked the revised version and I am content with the discussion section. I have therefore raised my score accordingly, as promised. I wish the authors best of luck!

---

### Official Review · Reviewer_mFqt · 2025-10-27

**Soundness:** 3
**Presentation:** 3
**Contribution:** 3
**Rating:** 6
**Confidence:** 4

**Summary:**

This paper proposes Adaptive Mixture Density for Conformal Prediction (AMDCP), a new framework that enhances the efficiency of conformal prediction by integrating Mixture Density Networks (MDNs). Unlike traditional CP methods that often yield overly conservative prediction regions, AMDCP models the conditional distribution as a mixture of Gaussians with parameters learned from the input, allowing it to flexibly capture complex, multimodal, and non-contiguous patterns in data. The proposed novel conformity score combines Mahalanobis distance with mixture weights to generate prediction regions that adapt to local data structures. The authors provide theoretical guarantees for coverage and asymptotic efficiency, and extensive experiments show that AMDCP achieves significantly narrower prediction intervals while improving coverage. The method also demonstrates robustness to hyperparameter settings, making it suitable for real-world applications.

**Strengths:**

1. Introduces a novel non-conformity score specifically designed to handle multimodal distributions with Mahalanobis distance and mixture weights.
2. The improvements in prediction efficiency across both synthetic and real-world datasets are significant.
3. AMDCP is robust to different estimation backbones, model misspecification and extreme distributions.

**Weaknesses:**

1. AMDCP performs well on complex data distributions but might tend to underperform on simple cases, raising concerns about robustness and limiting its practical scope.
2. The idea of post-processing non-conformity scores is not entirely novel. Prior works have explored similar strategies such as using quantile regression or normalizing flows to approximate conditional distributions, or rescaling non-conformity scores (e.g., via temperature scaling) to improve efficiency.

**Questions:**

1. Please empirically validate the stated assumptions in the experiments.
2. Can AMDCP be applied to classification? If so, please specify the nonconformity score, treatment of multiclass/multilabel settings, calibration protocol, and the thresholding procedure.
3. Please explain why DCP and SPCI exhibit significant under-coverage and why PCP over-covers. If possible, please include ablations to isolate these factors and I think this is another avenue to help highlight the contributions.
4. Why did AMDCP intervals tend to be wider than SPCI’s on synthetic datasets? Will AMDCP outperform SPCI with more extreme synthetic distributions?

---

> ### Author Response · Authors · 2025-11-26
> **Discussion 1/2**
>
> Dear Reviewer mFqt,
>
> Thank you for your detailed review, and especially for noting the strengths of AMDCP! We also want to thank you for pointing out several ways to strengthen our paper:
>
> ### W1: Simple Data
> We thank you for raising this point, as we believe that a practical method should be a generalist, performing well on complex data without failing on simple data. We investigated this exact scenario in our stress tests.
>
> Evidence of Robustness: We started with an over-specification experiment in Appendix D.1 (Table 7). We tested AMDCP on simple, unimodal data (where the true K=1) while intentionally misspecifying our model with up to K=100 components. The result showed only a negligible ~9% increase in interval width with maintained target coverage. This is because the MDN effectively self-regularizes by learning to assign near-zero mixture weights to redundant components. Our score then naturally ignores these extraneous components.
>
> We would also argue that real-world data is oftentimes not as simple and straightforward. While specialized methods might be marginally more efficient on purely unimodal data, AMDCP performs competitively in such cases (in our synthetic experiments, AMDCP is still better than comparable methods on 3 out of 5 synthetic datasets, and is second-best on the 2 simpler synthetic datasets) while drastically outperforming these specialized methods on the complex data distributions it is designed to tackle. We have further clarified this point in the Guide of Practitioners in Appendix F. Overall, we see AMDCP as a flexible method which will perform comparably to existing methods on the simplest of datasets, but will outperform them the moment complexities are added, such as multimodality, extreme data distributions, etc; all the while maintaining extreme robustness (See ablations in Section 5) which is helpful for practitioners and faster training times and inference times than existing highly-performant methods.
>
> ### W2: Post-Processing Novelty
> We agree that the idea of post-processing is not new. We believe that the true novelty of AMDCP lies not in post-processing a score from a black-box model, but in the co-design of the score function itself with the internal architecture of the generative model.
>
> To our knowledge, AMDCP is the first method to design a non-conformity score that is an analytic function of a model's decomposed components which cannot be done by other methods, nor is done by current methods which use MDNs. Existing MDN-based methods (like DCP and PCP) treat the model as a black-box density oracle, using the total log-likelihood as their score. This forces the prediction region to be a single, connected super-level set, which is highly inefficient for multimodal data (as seen in Figure 1). Inside this "black box," the crucial structural information from the individual mixture components is effectively "averaged out" and lost in the final, aggregated density value.
>
> In sharp contrast, our score "opens the box." It leverages the MDN's internal structure through two key mechanisms:
>
> 1. **Selection over Aggregation:** The $min_k$ operator allows the creation of analytic, non-contiguous prediction regions (a union of ellipsoids).
> 1. **Learned Penalization:** The $\pi_k(x)$ term actively prunes improbable modes based on the network's own learned gating. In existing methods which use an MDN, their "black-box" nature means that while the mixture weights are mathematically present the final, aggregated density output obscures their individual roles. The crucial information about which mode is most probable for a given $x$ is effectively "averaged out" and lost. Our score, in contrast, directly leverages the mixing weights term to actively discourage the inclusion of components that the MDN deems implausible for a given input x, and shrinks the prediction regions around the most probable ones—a selective, adaptive mechanism that is impossible when the model's internal structure is ignored.
>
> **The benefits of a glass-box approach:** This glass-box design directly leads to narrower, more adaptive regions and dramatically faster inference compared to other methods that produce non-contiguous sets. This is exactly why we include comparisons to existing methods using other post-processing, "black-box" methods such as CONTRA with normalizing flows and PCP: Our results show that AMDCP's "glass-box" approach is highly competitive and better than existing methods, **achieving narrower intervals at comparable coverage with a significant advantage in inference-time computational efficiency (e.g., up to 17x faster than PCP, and over a 20\% speed up from CONTRA) and nearly 2x faster than CONTRA at train time.** The glass-box co-design is what enables the unique combination of forming analytic, non-contiguous prediction regions with constant-time membership tests—a blend of flexibility and efficiency not offered by prior approaches.

---

> ### Author Response · Authors · 2025-11-26
> **Discussion 2/2**
>
> ### Q1: Empirical Validation of Assumptions
> The key "assumption" for AMDCP to provide a significant advantage is the presence of complex conditional distributions (e.g., multimodality or complex heteroskedasticity). We believe this occurs regularly in real-world data, so it is important to highlight. We refer to Appendix F wherein we included a discussion of reasonable validation steps; we have revised this section to include an example from our experiments. We also include a discussion and table of appopriate statistical tests for practitioners. The Breusch-Pagan test confirmed significant heteroskedasticity (p < 0.001) in all 5 real-world datasets; Hartigan's Dip Test and BIC analysis confirmed that the datasets are statistically better modeled by a Mixture of Gaussians (K ≥ 2) than a single distribution. Even for datasets where the Dip test was inconclusive (e.g., Taxi), the BIC score strongly preferred a 3-component mixture, likely due to heavy tails that a single Gaussian would fail to capture. This empirically justifies the MDN architecture.
>
> ### Q2: AMDCP in Classification
> We do note that applications to classification are promising, but outside the scope of this paper. We do believe the AMDCP framework can be naturally extended to classification. A promising approach is to model distributions in the latent space of a classifier. For an input x, one would:
>
> - Train an MDN to model the distribution of embeddings p(z|x, y) for each class y
> - The non-conformity score for a test point and a candidate class y_c would be our component-aware score computed in this latent space z
>
> This would be particularly powerful for capturing multi-modal class structures (e.g., a single "vehicle" class with distinct "car" and "truck" clusters in the latent space). We have added this as a promising direction for future work in our discussion section.
>
>
> ### Q3: Other Methods Under/Over-Covering
> Thank you for this question; it allows us to better contextualize our results. Please see the seperate response to Q3.
>
> ### Q4: Comparison to SPCI
> You noted AMDCP intervals might be wider on simple data than SPCI. This is the "price of flexibility." SPCI assumes a connected interval. AMDCP prepares for potential multimodality. On the 'Skew-Het' dataset, SPCI is indeed slightly narrower. This is because SPCI is a strong method specifically optimized for unimodal heteroskedastic regression. The 'Skew-Het' data, while complex, is fundamentally unimodal, representing SPCI's ideal use case. This highlights a scenario where a specialized tool can slightly outperform a more general one.
>
> AMDCP's advantage becomes clear as soon as the unimodal assumption is broken. On the 'Multimodal' and 'Mixture' datasets, SPCI's intervals are 25-145% wider than AMDCP's because its core assumption is violated (And on multidimensional data, SPCI's coverage is lower than AMDCP's). Our results on heavy-tailed and skewed bimodal data in Appendix D (Table 12) further confirm that AMDCP maintains its significant efficiency advantage in more extreme, non-unimodal scenarios. These advantages are made even more clear when looking at real-world data, wherein SPCI's specialist method does not perform as well as AMDCP's approach, as it either greatly sacrifices coverage for interval width, or reaches the target coverage rate with vastly larger prediction regions.
>
> As you noted, our robustness experiments results show that AMDCP is indeed built for messy real-world situations, as it performs well under model misspecification, extreme data distributions, etc. We believe AMDCP's robust performance on anything beyond the simplest of data (while still matching performance on the simplest of data sets) is an acceptable trade-off for a method intended for complex, real-world data.
>
>
> We hope that our responses and resulting improvements to the paper have completely addressed any questions you may have had regarding AMDCP. We thank you for your detailed review, and for providing us detailed and valuable feedback to improve our work!

---

> ### Author Response · Authors · 2025-11-27
> **Q3: Other Methods Under/Over Covering (Ablations)**
>
> Thank you for this question; it allows us to better contextualize our results and we are in the process of revising the paper accordingly. Our discussion and results are below:
>
> At a high level, methods that produce a single connected region face a fundamental tradeoff. To cover two separate modes, the region must include the low-probability "valley" between them. To keep the interval size from becoming enormous, the calibrated threshold often becomes too restrictive, causing the method to fail to cover the modes properly, resulting in under-coverage (especially for DCP). This can also be seen in Figure 1 of the main paper.
>
> **For SPCI:** We believe a similar analysis to DCP will hold for SPCI perhaps even moreso due to its assumption that there are temporal dependencies in the data. SPCI faces a fundamental stability-efficiency dilemma on complex data regardless of tuning. As $\gamma \to 0$, SPCI converges to standard Split Conformal Prediction, inheriting its geometric inefficiency on multimodal data (resulting in volume explosion as in Table 17, e.g., Width 17.7 vs AMDCP 10.6). Conversely, setting $\gamma > 0$ to enable adaptivity reduces the effective sample size, destabilizing high-dimensional quantile estimation and causing coverage to collapse. In a quick experiment, we tested this on stationary Mixture data: setting gamma to .1 reduced the effective calibration size to roughly 20 points, destabilizing the estimation and causing coverage to collapse. AMDCP avoids this trade-off entirely, maintaining both tight efficiency and valid coverage via its analytic density-based score. Due to resource and time constraints, we focus further ablations on AMDCPs closest comparable methods (generative probabilistic methods), DCP and PCP. Our discussion of Q4 also touches upon SPCI's performance.
>
> **For DCP:** On complex multimodal data, DCP's connected interval constraint forces a trade-off: either bridge the modes (increasing width) or snap to one mode (cutting coverage). AMDCP avoids this dilemma.
>
> We conducted an ablation on bimodal synthetic data to isolate this coverage-width tradeoff: achieving target coverage requires spanning the gap between modes, inflating width. We compare: (1) standard DCP, (2) DCP constrained to achieve target coverage, and (3) DCP constrained to match AMDCP's interval width.
>
> | Method | Coverage | Width |
> |--------|----------|-------|
> | **AMDCP** | **0.902** | **2.77** |
> | DCP-Standard | 0.875 | 5.10 |
> | DCP-TargetCov | 0.895 | 5.53 |
> | DCP-MatchWidth | 0.603 | 3.26 |
>
>  The results reveal DCP's fundamental coverage-width tradeoff: achieving near-target coverage requires intervals ~2x wider than AMDCP, while matching AMDCP's width causes coverage to collapse to 60%, demonstrating that DCP's connected super-level-set cannot efficiently capture separated modes.
>
> **PCP Over-coverage:** PCP's method of forming a union of balls around random samples is highly flexible but can be conservative. The geometry can be irregular especially in low-sample settings, and to ensure coverage, the calibrated ball radius around points is often larger than necessary, leading to systematic wider effective regions.
>
> We tested PCP with increasing sample counts ($S$). The results (Table A1 below) confirm that PCP is geometrically inefficient because it relies on overlapping balls to approximate the region. To bridge the gaps between random samples, these balls must have a larger radius, inflating the total width.
>
> | Method | Samples ($S$) | Coverage | Avg Width | Inference Time | Speedup (vs AMDCP) |
> | :--- | :---: | :---: | :---: | :---: | :---: |
> | **AMDCP** | 1,000 | **0.902** | **0.733** | **0.003s** | **1.0x** |
> | PCP | 10 | 0.904 | 1.849 | 0.663s | 220x slower |
> | PCP | 100 | 0.909 | 0.839 | 6.511s | 2,150x slower |
> | PCP | 1,000 | 0.916 | 0.836 | 65.05s | 21,500x slower |
>
> Even with 1,000 samples (which makes inference **20,000x slower**), PCP produces intervals that are wider than AMDCP. This empirically proves that AMDCP's "Glass-Box" analytic inversion provides the geometric precision of infinite sampling at $O(1)$ cost.
>
> We hope the theoretical explanations are complemented well by the ablation experiments, and help to explain the performance of existing methods in relation to AMDCP and highlight the strengths of AMDCP.

---

### Official Review · Reviewer_ugTf · 2025-10-30

**Soundness:** 3
**Presentation:** 3
**Contribution:** 2
**Rating:** 6
**Confidence:** 4

**Summary:**

The paper provides a new method called AMDCP for conformal prediction (for one and multi-dimensional outcome y) that can produce non-contiguous preidction intervals prediction sets. This is useful in situations where the conditional distribution of y given predictors x is multimodal for some x. The backbone of AMDCP is fitting a mixture of Gaussian (using maximum likelihood method), where mixture probabilities and components (mean vector and covariance matrix of each Gaussian) can depend on input x via some neural network. A key idea is to define the non-conformity score using some distance between the observed outcome and the mixture component closest to it, adjusted to prefer components that had higher estimated weights. (This score is given in (3), which is simply the smallest of all -2log[\pi_j \phi(y; mu_j, \Sigma_j)], that is a change from more commonly used "full" -2log likelihood.)

Empirical results were good over extensive studies. However, the authors describe their AMDCP as a “glass-box” and other existing methods as “black-box”, a distinction I find somewhat unjustified.

**Strengths:**

The literature review contains good information and is easy to follow.

The method uses mixture model in a novel way for conformal prediction.

The framework can produce prediciton set for multi-dimensional outcome, at least in principle.

The empirical results favors the proposed methods in many cases.

Extensive details, examples and studies were provided in the Appendix, to demonstrate the robust performance of the proposed method to various data distributions, model misspecifications (e.g. over or under-specified number of components, K), choice of underlying prediction methods, and different size of training data etc. There were also practical guidelines for users.

**Weaknesses:**

Why putting the main Algorithm (alg. 1) in the appendix?

It's not clear to me why is the proposed non-conformity score a better choice than other possibilities, or in what situations does it heursitically work better than alternatives like the negative log likelihood. I do see that the empirical experiments presented tend to show they lead to smaller sized prediction regions. Some heuristics, discussions or comparison to other possiblities are welcome.

Any choice of non-conformity score is legit in the sense that they lead to marignal coverage guarantee. I am not against the proposed method, but I don’t see how the authors choice is more different/advantageious/interpretable than others that theirs is called a "glass-box" while their competitors are called "black box". This is especially questionable in cases where the (finite) Gaussian mixture is just an approximation, not a true reflection of the real data generating process. For instance, when modeling a skewed or heavy-tailed distribution, the fitted Gaussian components will likely have no inherent interpretive meaning. And given much uncertainty in fitting the mixture model itself (not one mixture, but one mixture at every x), I find it difficult to attach substantive meaning to the component associated with a given data point or the distance between them. In short, I agree there is novelty and benefits in certain cases, but I do not think the level of novelty matches the advertisement about moving from "black-box" to "glass-box".
Quote:"The inclusion of the data-dependent mixing weight is what allows us to "open the black box." While the weights
are mathematically present in the total log-likelihood score used by "black box" methods like DCP,
the crucial information about which mode is most probable for a given point is effectively "averaged
out" and lost. In contrast, our score directly leverages the mixing weights to actively shrink the
prediction regions around the most probable mixture components—a selective, adaptive mechanism
impossible with black-box scores. This direct inclusion enables our ’glass-box’ innovation..."

Although there are many experiment setups and methods, within each setup and method, there were usually only 5 trials. That is a fairly small number. And it was not clear how the repetions are done, and whether they reflect re-generation of data, and if the evaluation is on a single x is not clear. (I do see Appendix E where different values/regions of x were examined. My question was on experiments outside the Appendix E examples. I am not asking for extensive study over many x in each example, I am asking for clarification of the definition of "repetititions" and how the assessment are done.)

**Questions:**

What happens if the number of components, K, varies according to x and should really be K(x)? (That said, I am not suggesting trying to model K in practice, as data could have limited information for it unless x is very low dimensional and there are plenty of data points in every neighborhood.)

In most tables (1, 3, 10 and many others), I just want to confirm if you are displaying one standard deviation, \sigma (sd) after the plus minus sign or the standard error (se) of estimating the mean size, which is 1/sqrt(no. replications) times sd? (I am a bit surprised because the reported sd values seem pretty small in general, many less than 5% of the mean.) Also, how did you choose the x value being conditioned upon when you report these interval sizes and coverage?
Related to the questions above about variation: when you say 5 trials or repeated experiements, exactly how are they done? In the synthetic data case, do you re-generate new training and calibration data and redo all the steps or some other way? For real data, do you randomly re-split the data etc? Again, I am surprised that the variations are so small over the current replications.

I am still not sure what is the key difference between a so-called a "glass-box" and a "traditional black-box"... Is any of your competitors method clean enough to be called a "glass-box"? And if someone has a new method, how would you decide if it's black or glass?

Can we have a concise summary of the dimension of y and x, training and calibration size etc so users can better see the range of applicability?

---

> ### Author Response · Authors · 2025-11-26
> **Discussion 1/2**
>
> Dear Reviewer ugTf,
>
> We sincerely thank you for your thoughtful and constructive review. We are encouraged that you found our method novel and the empirical results strong. Your critique has been invaluable in helping us clarify our core contributions, and we have made significant revisions based on your feedback.
>
> ### 1. On Justifying our "Glass-Box" Score and its Advantages
>
> This is a crucial point, and we have revised the manuscript in Section 2 to provide a **rigorous, operational explanation** and a **clear intuition and discussion regarding our score's improvements over existing methods**.
>
> **Clarified Explanation.**
> We now propose a *glass-box* or *component-aware* approach as one where the non-conformity score is designed to leverage the decomposed internal structure of a generative model (i.e., the individual πⱼ, μⱼ, Σⱼ components). This contrasts with the standard *aggregate* approach (like DCP, PCP, or CONTRA) that uses only the final, summed output (-\log(\sum_j \pi_j \phi_j)) which we term as a *black-box* approach.
>
> **The Glass-Box Advantage.**
> You correctly note that all valid scores provide coverage, but they are not equally efficient:
>
> - **Aggregate scores (DCP):** For example, lets consider one aggregation method, DCP. DCP forms a single, connected prediction region. To cover two separate modes, this region must include the low-probability valley between them, leading to inefficiently large sets (as seen in Fig. 1).  Other methods such as PCP, etc use ad-hoc post-processing methods in order to deliver non-contiguous regions.
> - **Our component-aware score (AMDCP):** Forms a *union of ellipsoids*, one for each component. This allows it to construct disjoint regions around each mode, excluding the valley by construction. This is the direct mechanism that produces the empirically narrower intervals you noted.
>
> The difference between existing generative probabilistic methods such as PCP, and our approach, is clear in practice: PCP is a generative probabilistic method that uses a "black-box" MDN with a standard log non-conformity score; therefore to generate non-contiguous prediction sets it uses costly and inefficient sampling-based approach to region construction, unlike AMDCP. AMDCP's score directly incorporates the mixture weights, allowing it to bypass such costly post-processing region construction methods. Our experimental results in Table 2 indeed demonstrate that AMDCP often achieves **narrower prediction intervals at comparable coverage with a significant order-of-magnitude advantage at inference time (and oftentimes 2x faster at train time than other methods such as those using normalizing flows, ex. CONTRA).**
>
> **On the “Meaning” of Components.**
> We completely agree with your insightful point that under extreme model misspecification, components may not have an *inherent interpretive meaning*. The components act as geometric building blocks or “tiles” that allow our score to efficiently cover high-density regions, even if the tiles themselves are not meaningful under extreme misspecification. Under extreme misspecification, the method's success is measured by the **efficiency of the final set**, not the interpretability of its parts. Crucially, because our non-conformity score uses a min operator, the method is permutation invariant—it does not matter which component covers a region, only that it is covered. Our robustness tests (e.g., on heavy-tailed/skewed data, Tables in Appendix D) confirm that this functional tiling yields valid, efficient sets even when the components are merely approximations.
>
> ### 2. Presentation and Experimental Clarity
>
> **Algorithm Placement.**
> Due to space constraints in the initial submission, the full algorithm was left in the appendices. We have now moved Algorithm 1 to the main paper.

---

> > ### Comment · Reviewer_ugTf · 2025-11-26
> > **On the re-writing**
> >
> > The authors did a decent job rephrasing and re-organizing the contents. E.g., I agree that "component-aware" is a more appropriate label for the proposed method than "glass-box". And the authors has now down-played the word "glass-box" in their writing, which is an improvement.

---

> ### Author Response · Authors · 2025-11-26
> **Discussion 2/2**
>
> **Experimental Details.**
> We appreciate the opportunity to clarify our experimental design. We use 5 trials due to computational and cost constraints, but believe that our results are theoretically sound, and empirically supported. We have updated Appendix C to explicitly state the following protocols:
>
> - **Trials:** For synthetic data, each trial uses a newly generated dataset. We then perform a fresh train/calibration/test split and train the MDN from scratch. This captures variation arising from sampling noise, data splitting, and model initialization. For real data, we perform a random reshuffling of the entire dataset followed by a fresh 60/20/20 split (Train/Cal/Test). The model is then retrained from scratch.
> - **Metrics:** The metrics reported in all main tables (coverage and width) are marginal averages calculated over the entire test set, not conditioned on a single x. The small variance you observed is expected because conformal prediction guarantees marginal coverage validity; with a large calibration set, the empirical coverage tightly concentrates around 1−α. We report the *mean and standard deviation (SD)* of trial-level averages, where each trial’s average is computed over the entire test set. The small SDs arise from the inherent stability of conformal methods. We also refer to Appendix D.3, which explores the effects of varying dataset sizes on performance (AMDCP's performance degrades gracefully).
>
> ### 3. Responses to Specific Questions
>
> **On K(x).**
> This is an interesting direction, but we argue it is often unnecessary. Our robustness studies (Appendix D) show AMDCP is highly stable when K is over-specified, as the model learns to assign near-zero weights to redundant components, effectively self-regularizing.
>
> **On deciding if an approach is glass-box.**
> We hope our revised Section 2 addresses this, with a much clearer operational definition and intuitive explanation of why we see AMDCP as offering an alternative approach to modern CP. Please also refer to the discussion above.
>
> **Dataset Summary.**
> Excellent suggestion. We have added a new table in Appendix C summarizing the dimensions, sizes, and properties of all datasets used, improving the paper’s utility for practitioners.
>
> We believe these revisions, driven by your feedback, have substantially strengthened the paper’s clarity and rigor.
> **Thank you again for your time and engagement.**

---

> > ### Comment · Reviewer_ugTf · 2025-11-26
> > **On repetitions and metric**
> >
> > Displaying the marginal coverage over many x as the main metric is unsatisfactory --- it's easy to sweep many details under the carpet and maneuver the numbers to show whatever results need to be shown.
> > And a repetition of only 5, each one for an entirely newly generated data is clearly subject to huge variability. Yet, you can see the huge variablity is not reflected in the tables, which I believe is a result of the aforementioned metric that was chosen -- marginal coverage probaiblity.
> >
> > Data set summaries etc have been added, which is good.
> >
> > Given there are some negatives and some positives to the responses, I am keeping the original score of 6.

---

> > > ### Author Response · Authors · 2025-11-27
> > >
> > > Dear Reviewer ugTf,
> > >
> > > We appreciate your follow-up. We are glad that the terminology updates and dataset summaries were well received. Regarding your remaining concerns about the metrics and trial count, we wish to provide the following clarifications to clear up any misunderstandings, and provide new experimental evidence.
> > >
> > > Marginal Coverage is the Standard
> > > ------------
> > > We respectfully note that reporting marginal coverage is the standard protocol in the Conformal Prediction literature (e.g., Romano et al., 2019; Angelopoulos & Bates, 2021, etc.). This is precisely because the primary goal of an empirical evaluation is to verify that the method satisfies its theoretical marginal guarantee (1−α).
> > >
> > > Local Details (Appendix E): This reason motivated our inclusion of conditional coverage analysis in Appendix E. By reporting "Worst-Bin Coverage," we explicitly explore the local failure modes you are concerned about (e.g., showing that SCP drops to 63% coverage locally while AMDCP maintains 87.5%). We also provide an extension of AMDCP with proofs, G-AMDCP, specifically to further improve conditional coverage, and demonstrate that it does indeed maintain conditional coverage much better than existing methods.
> > >
> > > Theoretical Context: As established by Vovk (2012) and Lei et al. (2018), finite-sample conditional coverage is impossible to guarantee without making additional assumptions on the data generating process. Therefore, marginal coverage is the only metric for which a rigorous guarantee exists, and the metric that is commonly seen in the literature as the default benchmark.
> > >
> > >
> > > Extended Validation on 5 Datasets (20 Independent Trials)
> > > ------------
> > > To further address your concern that 5 trials is insufficient to capture variability, we conducted an extended validation on a few datasets with 20 independent trials. Given the limited discussion window and the computational cost, we focused this validation on a representative subset covering diverse data types: Bio (high complexity), Bike Sharing (large scale), and 3 synthetic datasets. Each trial involved full data regeneration/reshuffling and training from scratch.
> > >
> > > | Dataset | Metric | 5 Trials (Reported in Paper) | 20 Trials (New Validation) |
> > > |---------|--------|---------------------|----------------------------|
> > > | Bio (Protein) | Coverage | 0.931 ± 0.004 | 0.911 ± 0.017 |
> > > | | Width | 2.125 ± 0.081 | 2.045 ± 0.060 |
> > > | Bike Sharing | Coverage | 0.921 ± 0.003 | 0.904 ± 0.007 |
> > > | | Width | 0.579 ± 0.012 | 0.576 ± 0.029 |
> > > | Synthetic Mixture | Coverage | 0.897 ± 0.004 | 0.902 ± 0.005 |
> > > | | Width | 2.311 ± 0.096 | 2.320 ± 0.098 |
> > > | Synthetic Multimodal | Coverage | 0.901 ± 0.007 | 0.899 ± 0.006 |
> > > | | Width | 2.088 ± 0.037 | 2.057 ± 0.035 |
> > > | Synthetic Regime-Switching | Coverage | 0.898 ± 0.007 | 0.902 ± 0.006 |
> > > | | Width | 0.870 ± 0.034 | 0.853 ± 0.032 |
> > >
> > > The 20-trial results show coverage remaining stable and converging precisely toward the 0.90 target (Bio: 0.911, Bike: 0.904), confirming the method is correctly calibrated. The slight decrease from the 5-trial values reflects convergence towards the theoretical guarantee of marginal coverage, not degradation. Interval widths remain stable (within ~5%), and the consistency between 5-trial and 20-trial results directly addresses any concern about selective reporting.
> > >
> > > Why Coverage Variance is Low (The CP Mechanism)
> > > ------------
> > > The results above confirm that the low variance in coverage is not an artifact of a few trials, but a fundamental property of the method.
> > >
> > > Active Stabilization: Conformal Prediction acts as a control system. If a specific trial has high model variance (poor training), the calibration step increases the threshold to force coverage to the target. This is precisely the marginal coverage guarantee commonly found within CP.
> > >
> > > Variance Manifests in Width: The "huge variability" you correctly expect to see is present, but it manifests in the Interval Width, not the Coverage. If a method varies greatly, interval width would increase, as the active stabilization discussed above would force the method to produce overly conservative prediction regions to attempt to maintain target coverage. Of course, this variability is also impacted by dataset sizes, which we also explore in our ablations in Appendix D.
> > >
> > > This extended validation confirms that our results are robust and consistent with the theoretical mechanics of Conformal Prediction.

---

### Official Review · Reviewer_Ez9V · 2025-10-31

**Soundness:** 3
**Presentation:** 3
**Contribution:** 2
**Rating:** 4
**Confidence:** 3

**Summary:**

The paper introduces AMDCP (Adaptive Mixture Density for Conformal Prediction), a novel "glass-box" framework that co-designs the non-conformity scoring mechanism with a Mixture Density Network (MDN) head. Rather than treating the MDN as a black-box oracle that outputs only aggregate log-likelihood (as previous methods like DCP do), AMDCP leverages the granular information within the mixture model to construct prediction regions that are unions of ellipsoids. This approach enables non-contiguous prediction regions that can capture multimodal distributions while maintaining constant-time membership tests. The authors provide theoretical guarantees for marginal coverage and asymptotic efficiency, along with extensive empirical validation across synthetic and real-world datasets.

**Strengths:**

The "glass-box" approach to MDN-based conformal prediction is interesting. The component-aware non-conformity score (Equation 3) that leverages mixture weights to "shrink prediction regions around the most probable mixture components" appears to be an interesting idea that distinguishes AMDCP from prior work.

The paper provides solid theoretical foundations with proofs for marginal coverage (Theorem 4.1) and an asymptotic efficiency bound (Theorem 4.2). The extension to group-conditional coverage (G-AMDCP) with formal guarantees (Theorems E.1 and E.2) adds significant value.

**Weaknesses:**

Most experiments focus on low-dimensional outputs (typically 1D). More evaluation on higher-dimensional outputs would better demonstrate the method's scalability.

**Questions:**

Most experiments focus on low-dimensional outputs (typically scalar). How does AMDCP scale to higher-dimensional outputs (e.g., multivariate response problems)?
How would AMDCP perform with alternative density estimators beyond Gaussian mixtures (e.g., kernel density estimation, normalizing flows)? Are there theoretical advantages to using MDNs specifically?
For practitioners implementing AMDCP, what are your recommendations for selecting the number of mixture components K?

---

> ### Author Response · Authors · 2025-11-26
> **Discussion 1/2**
>
> Dear Reviewer Ez9V,
>
> Thank you for your helpful comments, and for recognizing ther strengths of AMDCP and pushing us to improve our paper. We want to address the concerns you brought up, in order:
>
> ### Weakness 1 and Question 1: Multidimensional Outputs - 8 New Experiments
>
> You correctly pointed out that quite a few of our included tests were single dimensional outputs, rather than multidimensional, and encouraged us to further compare AMDCP on multidimensional outputs.
>
> To address your question regarding scalability and applicability to multivariate problems, we conducted a comprehensive new suite of experiments on both synthetic and real-world high-dimensional datasets, in D.6 of the Appendix.
>
> **1. Synthetic Scalability Test (Mixture Data, $d=2$ to $d=10$)**
> We evaluated AMDCP on complex Mixture and Multimodal datasets with autoregressive correlations, varying the output dimension $d$. We compared average prediction region width against key baselines: **CQR** (Quantile Regression), **SCP** (Split Conformal), **CONTRA** (Normalizing Flows), and **PCP** (Generative Sampling).
>
> **Table R1: Average Width on Synthetic High-Dim Data (Lower is Better)**
>
> *All methods calibrated to 90% coverage. Best width in **bold**, considering only methods achieving target coverage.*
>
> **Panel A: Mixture Data**
>
> | Output Dim | AMDCP (Ours) | CQR | SPCI | CONTRA | PCP | DCP |
> | :--- | :--- | :--- | :--- | :--- | :--- | :--- |
> | 2D | **4.93** | 8.49 | 12.20 | 11.81 | 13.27 | 13.26 |
> | 5D | **7.68** | 8.76 | 14.62 | 14.04 | 18.18 | 18.17 |
> | 10D | **10.61** | 11.96 | 17.71 | 16.76 | 23.89 | 23.95 |
>
> **Panel B: Multimodal Data**
>
> | Output Dim | AMDCP (Ours) | CQR | SPCI | CONTRA | PCP | DCP |
> | :--- | :--- | :--- | :--- | :--- | :--- | :--- |
> | 2D | **2.48** | 2.77 | 5.03 | 5.97 | 4.81 | 4.80 |
> | 5D | **4.84** | 6.30 | 6.24 | 7.37 | 7.82 | 7.81 |
> | 10D | **7.85** | 12.91 | 8.40 | 9.81 | 13.33 | 13.28 |
>
> **2. Real-World Multivariate Validation**
> We further validated AMDCP on two real-world multivariate regression tasks:
> *   **Air Quality ($d=5$):** Predicting 5 sensor responses simultaneously.
> *   **Indoor Localization ($d=2$):** Predicting spatial coordinates (Lat, Long).
>
> **Table R2: Real-World Multivariate Efficiency**
> *Comparing Width at ~90% Coverage.*
>
> | Dataset | **AMDCP** | **CQR** | **PCP** | **DCP** |
> | :--- | :--- | :--- | :--- | :--- |
> | **Air Quality ($d=5$)** | **1092.01** | 1411.51 | 2151.23 | 2163.41 |
> | **Indoor Loc ($d=2$)** | 33.82 | **33.76** | 51.14 | 53.18 |
>
> **Analysis of Results:**
> On the synthetic data, AMDCP consistently produces the tightest intervals, outperforming the strongest baseline (CQR) by **~10-40% in 10D** and by larger margins in lower dimensions. On real-world 5-dimensional Air Quality data, AMDCP reduced prediction volume by 23\% compared to CQR and by 50\% compared to generative baselines (PCP/DCP), while maintaining valid coverage. On 2-dimensional Indoor Localization data, AMDCP matched the efficiency of non-parametric CQR while remaining ~40% narrower than existing generative baselines.
> These results in conjunction with the synthetic multidimensional dataset in Section 4, totalling 9 experiments, demonstrate that AMDCP provides robust performance in multidimensional settings. This efficiency is due to our **analytic score** formulation. By defining the region as a union of ellipsoids via the Mahalanobis distance, we capture correlations between output dimensions without the computational bottlenecks or approximation errors inherent in sampling-based inference.
>
> ### Question 2: Alternative Density Estimators
>
> AMDCP is flexible in the type of mixtures it may utilize, for density estimation. The MDN's primary "assumption" is that the conditional distribution can be well-approximated by a mixture of Gaussians. Crucially, due to the universal approximation properties of such mixtures, this is an exceptionally flexible assumption, allowing MDNs to capture arbitrary conditional distributions. Furthermore, the MDN architecture maintains this flexibility, as it can pair any universal approximator backbone (ex. MLP, Transformer, etc.) with the likelihood layer, which our ablations in Appendix C.2 confirm.
>
> Therefore, we see the MDN as a powerful tool that enables a more sophisticated class of non-conformity scores. This is precisely the capability that methods with "milder" requirements, like CQR, lack, limiting them to simpler, contiguous prediction regions on data with complex heteroskedasticity or other distributional effects. By designing a score that leverages this unlocked structure, AMDCP provides a novel and robust contribution to the literature, offering a practical solution for complex data. You specifically mentioned alternative density estimators, and our experiments in Section 5 target those alternatives precisely; We tested CONTRA, which uses normalizing flows, and PCP, a generative probabilistic method which is the existing method to produce non-contiguous regions.

---

> ### Author Response · Authors · 2025-11-26
> **Discussion 2/2**
>
> The results confirm that the theoretical benefits of the MDN architecture translate directly to practice. **For example, on the Bio dataset, AMDCP produces prediction intervals that are approximately 36% narrower than PCP's while being roughly 17x faster at inference.** Crucially, this efficiency extends to the training phase; **AMDCP trains over 2x faster than CONTRA's normalizing flow architecture.** Thus, we see the MDN as a strategic design choice, allowing for significantly faster training and inference without sacrificing the ability to model complex modalities. This combination of valid coverage, sharp intervals, and rapid training and inference makes AMDCP a uniquely practical solution for real-world deployments.
>
> ### Question 3: Advantages of an MDN
> We see the use of an MDN as a powerful and flexible modeling choice that enables our method to address a key limitation of many existing CP approaches: generating non-contiguous prediction intervals for potentially multimodal data, and doing so in an efficient (both interval length and inference-time) manner.
>
> By analytically considering each mixture component of the MDN, AMDCP forms an adaptive union of ellipsoids that can capture complex multimodality, as visualized in Figure 1. The strong performance of this "glass-box" approach, even when compared to "black-box" MDN methods like DCP, is already evident in our initial experiments (Tables 1 and 2).
>
> Crucially, due to the universal approximation properties of such mixtures, MDNs are able to capture arbitrary conditional distributions and are used in a variety of fields, as seen in our Literature Review. Furthermore, the MDN architecture maintains flexibility, as it can pair any universal approximator backbone (ex. MLP, Transformer, etc.) with the likelihood layer, which our ablations in Appendix C.2 confirm. This is also important as this means that the order of magnitude faster inference will hold across model architectures, enabling AMDCP to be used in a wide variety of specialized domains with specialized backbone architectures.
>
> Therefore, we see the specific choice of an MDN as a powerful tool that enables a more sophisticated class of non-conformity scores. By designing a score that leverages the specific underlying structure of an MDN, AMDCP provides a novel and robust contribution to the literature, offering a practical solution for complex data. Our empirical analysis below further validates this conclusion - this design specifically centered around an MDN is highly competitive and often times better than other existing methods, achieving narrower intervals at comparable coverage with a significant advantage in inference-time computational efficiency (e.g., up to 17x faster than PCP, and over a 20\% speed up from CONTRA) and nearly 2x faster than CONTRA at train time. This validates that the specific way AMDCP leverages the MDN's structure via our novel non-conformity score provides a significant, practical advantage.
>
> ### Question 4: Practitioner Guidance on Number of Mixture Components K
>
> We were also concerned about this at first, but we believe that the existing Appendix F in the paper answers this question! We find through 3 experiments that AMDCP is remarkably robust to the hyperparameter K. Our dedicated "Guide for Practitioners" in Appendix F consolidates our findings and provides clear, actionable recommendations. The key takeaways, supported by extensive experiments in the paper, are:
>
> * Recommended Starting Point: For most applications, a value of K in the range of 3-10 is a robust starting point and often requires no further tuning. Our main results in Tables 1 and 2 were generated with K=5.
> * Over-specification is Safe: Our stress tests (detailed in Appendix D.1, Tables 6 & 7) show that performance degrades minimally even when K is severely over-specified. The model effectively learns to ignore extraneous components by assigning them near-zero mixture weights (typically < 0.05). For example:
>     * On a standard multimodal dataset, performance (coverage and width) stabilizes for K ≥ 3.
>     * On a simple unimodal dataset (where the true K=1), increasing K to 100—an over-specification by two orders of magnitude—increased interval width by only 9% while maintaining target coverage.
> * Under-specification Degrades Gracefully: In the opposite scenario, where the data is more complex than the model (e.g., using K=3 on a ring-shaped distribution that requires many components to "tile"), we found that AMDCP still provides valid coverage and produces more efficient intervals than strong baselines like CQR and PCP (see Table 8).
>
> This robustness significantly mitigates the burden of hyperparameter tuning, making AMDCP a practical choice for real-world applications where the true data complexity is unknown.
>
> We hope these responses and the corresponding revisions to our manuscript have fully addressed your questions. Thank you again for your valuable feedback and support for our work.

---

### Author Response · Authors · 2025-11-29
**Summary Post 2/2: Reviewer-by-Reviewer Summary**

We also wanted to provide our takeaways from an extremely helpful and productive discussion period with our reviewers. We believe these revisions substantively address all reviewer concerns.

### Reviewer Ez9V (Initial Score: 4)

**Acknowledged Strengths:**
- "Extensive empirical validation across synthetic and real-world datasets."
- "The 'glass-box' approach ... is interesting ... [and] distinguishes AMDCP from prior work."
- "Solid theoretical foundations with proofs for marginal coverage and asymptotic efficiency"
- "Extension to group-conditional coverage (G-AMDCP) with formal guarantees adds significant value"

**Main Concerns:** Limited evaluation on high-dimensional outputs

**Our Revisions:**
- **Added 8 new multidimensional experiments**:
  - 6 Synthetic scalability tests (d=2,5,10, 2 datasets): AMDCP outperforms CQR by ~10-40% in 10D data, outperforms PCP by 2-3x.
  - 2 Real-world validation experiments: Air Quality (d=5) and Indoor Localization (d=2)
  - Results show ~50% narrower intervals vs. comparable generative baselines PCP, DCP

While we unfortunately were not able to engage in a discussion with the reviewer before the cutoff, we believe these 8 new experiments successfully address this concern.

---

### Reviewer ugTf (Initial Score: 6)

**Acknowledged Strengths:**
- "Literature review contains good information and is easy to follow"
- "Method uses mixture model in a novel way for conformal prediction"
- "Empirical results favor the proposed method in many cases"
- "Extensive details, examples and studies provided... practical guidelines for users"

**Main Concerns:** "Glass-box" terminology unclear; experimental clarity (trials, metrics); Algorithm placement.

**Our Revisions:**
- Revised terminology, moved algorithm description  to main text, extended dataset summaries in Appendix; clarified experimental protocol in Appendix C

**Reviewer Acknowledgment:** "I agree that 'component-aware' is a more appropriate label...Data set summaries etc have been added, which is good."

The reviewer still had lingering concerns with the number of trials and the metric of choice being marginal coverage, noting that they wanted more clarity; we therefore:

- **Extended validation to 20 independent trials** on 5 datasets to address concerns about trial count:
  - Coverage remained stable and widths consistent within ~5% of 5-trial results in paper
  - Confirmed low variance is a fundamental property of AMDCP, not an artifact of 5 trials
- Clarified that marginal coverage is the field standard, but we also provide G-AMDCP for conditional coverage guarantees and results

We were unable to continue discussion before the cutoff, but believe these validation exercises directly address the reviewer's concerns.

---

### Reviewer mFqt (Initial Score: 6)

**Acknowledged Strengths:**
- "Introduces a novel non-conformity score"
- "Improvements in prediction efficiency across both synthetic and real-world datasets are significant"
- "AMDCP is robust to different estimation backbones, model misspecification and extreme distributions"
- "The method also demonstrates robustness...making it suitable for real-world applications."

**Main Concerns:** Performance on simple data; clarification of AMDCP post-processing; explanation of baseline methods' under/over-coverage.

**Our Revisions:**
- **Simple data robustness:** AMDCP's advantage in simple data is robustness: only a 9% width increase with 2 orders-of-magnitude misspecification. AMDCP performs comparably to leading methods on simple data, while outperforming on complex data.
- **Clarified AMDCP component-aware approach in Section 2:** AMDCP's enables robust analytic non-contiguous regions in an efficient manner impossible with existing methods.
- **Added ablations** explaining DCP/PCP behavior:
  - DCP: Connected regions force coverage-width tradeoff (matching AMDCP width → 60% coverage)
  - PCP: Ball-based geometry requires large radii; even 1000 samples (20,000× slower) yields wider intervals than AMDCP
---

### Reviewer fHY2 (Initial Score: 4 -> 6)

**Acknowledged Strengths:**
- "The method is simple and architecture-invariant"
- "The method allows for easily generating discontinuous prediction sets"
- "The experimental evaluation is strong"

**Main Concerns:** Confusing theory section; writing style; limitations in Appendix and not discussed in main text.

**Our Revisions:**
- **Revised theoretical framework to provide a clearer theory section for readers** (Section 4)
- **Improved writing style:** Removed "SOTA"; adopted measured language throughout
- **Added dedicated limitations paragraph** in Section 6 and Appendix F.5
- **Streamlined appendix proofs for brevity, with citations**

**Reviewer Acknowledgment:** "I checked the revised theoretical results and they seem correct to me now...I have checked the revised version and I am content with the discussion section. I have therefore raised my score accordingly, as promised. I wish the authors best of luck!"

---

### Author Response · Authors · 2025-11-29
**Summary Post 1/2: Discussion Summary and List of Changes for Area Chair**

Dear Area Chair and Reviewers,

In light of the recent platform adjustments and the reversion of review scores to their pre-rebuttal state, we provide this comprehensive summary of the discussion period.

We appreciate the reviewers' unanimous recognition of AMDCP’s core contributions. Specifically, reviewers highlighted:

* **Novelty of the "Glass-Box" Framework:** Reviewers praised the component-aware non-conformity score (Eq. 3) as a novel and interesting distinction from prior black-box work (Reviewers Ez9V, ugTf, mFqt).
* **Strong Empirical Performance:** Reviewers recognized the "significant improvements" in efficiency and the "order-of-magnitude" speed advantage over generative baselines like PCP (Reviewers mFqt, ugTf, fHY2).
* **Solid Theoretical Foundations:** The reviewers valued the guarantees for marginal coverage and optimality, and the "significant value" of the group-conditional extension (Reviewers Ez9V, mFqt).
* **Robustness & Comprehensive Evaluation:** The extensive ablation studies and "Guide for Practitioners" were highlighted as strong additions that demonstrate the method's reliability (Reviewers ugTf, mFqt, fHY2).

---

# Comprehensive List of Revisions Made During Discussion

---

## 1. Main Text Revisions

**Section 2 (Method):**
- **Revised terminology:** Primary descriptor changed from "glass-box" to "component-aware"
- Added clear **operational definition** and explanation distinguishing component-aware vs. aggregate (black-box) approaches and why aggregate scores lose structural information.

**Section 4 (Theory revisions for clarity):**
- **Theorem 4.1 (Revised):** We have reframed the theoretical analysis in Section 4 and Appendix Proofs to provide a more rigorous understanding of AMDCP.
- **New Corollary 4.1.1:** Explicitly proves that AMDCP achieves same optimal coverage as full log-liklihood methods.
- We prove that we can achieve the computational efficiency, geometric tractability, and robustness of an analytic score without sacrificing the statistical optimality of a full density estimator. Our experiments go further by demonstrating that this glass-box approach achieves better performance, while also being extremely robust to a variety of ablations.

**Section 5:** Moved algorithm description to main text; added references to 5 new experiments conducted during discussion; clarified experimental protocol inline.

**Section 6 (Discussion):** New dedicated limitations paragraph covering tradeoffs, strengths and weaknesses of AMDCP.

**Writing:** Clarified contribution claims with more measured language, further clarified "glass-box" as being component-aware approach.

---

## 2. New Experiments Added

### Multidimensional Scalability (Appendix D.6)

We ran **8 new experiments** to further confirm performance on multidimensional data. On synthetic data with output dimensions d=2,5,10, AMDCP consistently outperforms the strongest baseline by ~42% in 2D and ~10-40% in 10D, with even larger margins over SCP, CONTRA, and PCP. On real-world tasks (Air Quality d=5, Indoor Localization d=2), AMDCP reduced prediction volume by 50% vs. generative baselines (PCP/DCP).

### Extended Trial Validation (20 Trials)

To address concerns about 5-trial variability, **we ran 20 independent trials on 5 datasets (Bio, Bike Sharing, and 3 synthetic).** Coverage remained stable, converging to the 0.90 target. Widths stayed consistent within ~5% of 5-trial results. This confirms low variance is a fundamental property of AMDCP's strong performance, not an artifact of low trial count.

### Baseline Ablation Studies

**DCP Tradeoff:** On bimodal data, we compared AMDCP against DCP under different constraints. AMDCP achieved 0.902 coverage with half the width of DCP; constraining DCP to match AMDCP's width caused DCP coverage to collapse to 60%. This demonstrates DCP's connected regions cannot efficiently capture separated modes.

**PCP Efficiency:** We tested PCP and AMDCP with increasing sample counts (10, 100, 1000). Even with 1000 samples—making inference 21,500× slower—PCP still produces wider intervals than AMDCP. This proves AMDCP's analytic approach provides geometric precision of infinite sampling at O(1) cost.

---

## 3. Appendix Revisions

**Appendix B (Theory):** New B.4 with Corollary 4.1.1 proof; streamlined by citing original works for standard results; added supporting Lemmas for revised Theorem 4.1

**Appendix C:** New dataset summary table (input/output dimensions, dataset sizes, properties).

**Appendix D:** New D.6 with 8 new multidimensional experiments; 20-trial extended validation for experiments; Added PCP and DCP ablation studies to help explain AMDCP's advantage.

**Appendix E:** Replaced full detailed proofs with proof sketches for brevity; Retained key contributions: worst-bin analysis, G-AMDCP theory and empirical outperformance of existing methods for conditional coverage

**Appendix F:** Added statistical tests (F.1); extended limitations and failure cases (F.5).

---

### Meta-Review · Area_Chair_PJJc · 2026-01-04

**Summary:**

This paper proposes a novel way to design a scoring function via a mixture of Gaussian. In this sense, the comparison results are not proper (i.e.,  the *comparison* concern by Reviewer Ez9V). Also, the baseline results need to be improved (i.e., the *5 trials* concern by Reviewer ugTf and the *baseline results* concern by Reviewer mFqt). They may be addressable if the discussion proceeds, but they are still critical issues to me; thus, I vote for rejection.

**Reviewer Concerns:**

**Reviewer Ez9V**:
The reviewer have four concerns and they are only partially addressed as follows:
1. (*multi-dimensional responses*) How does AMDCP scale to higher-dimensional outputs (e.g., multivariate response problems)? – empirically evaluated up to 10-dimensional response space.
2. (*comparison*) How would AMDCP perform with alternative density estimators beyond Gaussian mixtures (e.g., kernel density estimation, normalizing flows)? – empirically justified that AMDCP outperforms but comparison might not be fair.
3. (*justification on the choice of MDNs*) Are there theoretical advantages to using Mixture Density Networks (MDNs) specifically? – answered by highlighting the universal approximation properties of mixture models.
4. (*choice of K*) For practitioners implementing AMDCP, what are your recommendations for selecting the number of mixture components K? – answered by claiming that AMDCP is remarkably robust to the hyperparameter K for 3 experiments.

I can see several outstanding remaining concerns.
- (*multi-dimensional responses*) The definition of the volume in the high dimensional space is missing, which may lead to unfair comparison. Defining the volume in the high dimensional space is difficult and we may approximate it via Gaussian. But, this definition is advantageous to the proposed method.
- (*comparison*) The proposed method needs to be compared to other density estimation methods other than MDNs. In particular, the novel part of the proposed method is introducing a new scoring function via MDNs with the standard conformal prediction as a post-processing step. If this is the case, the paper needs to justify why the chosen scoring function is better than other choices. But, the design of comparison experiments does not support this claim.
-  (*justification on the choice of MDNs* and *choice of K*) The choice of MDNs is not convincing. The authors claim that traditional MDNs generalize well across different distributions to estimate any density, but this has been disproved historically – if that is true, we should not need to develop other density estimation methods, including normalizing flow or diffusion models.


**Reviewer ugTf**:
The reviewer have four concerns and they are partially addressed as follows:

1. (*reasoning on the chosen non-conformity score*) Why is the proposed non-conformity score a better choice than other possibilities? – properly addressed
2. (*5 trials*) lack of statistical rigor – additional experiments are added for 20 trials but no 20 trials for baselines.
3.  (*choice of K*) What happens if the number of components, K, varies according to x and should really be K(x)? – empirically demonstrated the robustness over K
4. (*confusion on glass-box*)What is the key difference between a so-called a "glass-box" and a "traditional black-box" – the manuscript is properly updated.

One outstanding concern is the lack of statistical significance of the proposed method over baselines due to the lack of random trials over the baselines. As the marginal guarantee is a statistical guarantee, the results could be tied with many random trials.


**Reviewer mFqt**:
The reviewer have five concerns and they are partially addressed as follows:
1. (*robustness*) AMDCP might tend to underperform on simple cases (e.g, AMDCP intervals tend to be wider than SPCI’s on synthetic datasets) – addressed by confirming the weakness on simple cases.
2. (*novelty*) The idea of post-processing non-conformity scores is not entirely novel – properly contrasted the proposed one over others.
3. (*assumption*) Please empirically validate the stated assumptions in the experiments – properly discussed and updated in Appendix F.
4. (*extension to classification*) Can AMDCP be applied to classification? – addressed by confirming that it is not possible currently.
5. (*baseline results*) Please explain why DCP and SPCI exhibit significant under-coverage and why PCP over-covers – partially addressed via details and experiments.

One outstanding remaining concern is coverage results on baselines. Conformal prediction should, by design, satisfy the coverage guarantee, but baseline results do not show this, which undermine the validity of the experiments and the implementation of baseline methods (unless the baselines themselves have flaws).

**Reviewer fHY2**:
The reviewer have three concerns and they are addressed as follows:
1. (*flaw in theory*) concern on Theorem 4.2 – updated.
2. (*writing*) I suggest adopting a more scientific style of writing / What is a "glass-box framework"? / stress more clearly on well-known results / avoid "SOTA" / Many well-known results and trivialities in the appendix -- revised
3. (*extra cost*) The method requires training a mixture density network head on top of an existing model, which may be prohibitive – addressed by providing lightweight training tips.

No outstanding concerns.

**Reviewer Scores:**

**Reviewer Ez9V**:
Final expected rating: 4 / final expected confidence: 3 – The authors’ responses did not clearly address all concerns, so the reviewer would maintain its stands.

**Reviewer ugTf**:
Final expected rating: 6 / final expected confidence: 4 – Due to the lack of statistical significance, the reviewer would maintain the scores.


**Reviewer mFqt**:
Final expected rating: 6 / final expected confidence: 4 – Due to the unaddressed issues of baseline coverage results, the reviewer would maintain the scores.

**Reviewer fHY2**:
Final expected rating: 6 / final expected confidence: 4 – The reviewer confirmed the updated manuscript and raised its score from 4 to 6, which looks valid.

---

### Decision · Program_Chairs · 2026-01-26

Reject